# Near-Optimal Regret for KL-Regularized Multi-Armed Bandits

**Kaixuan Ji** [* 1]  **Qingyue Zhao** [* 1]  **Heyang Zhao** [* 1]  **Qiwei Di** [1]  **Quanquan Gu** [1]

## Abstract

Recent studies have shown that reinforcement learning with KL-regularized objectives can enjoy *faster* rates of convergence or *logarithmic* regret, in contrast to the classical $\sqrt{T}$-type regret in the unregularized setting. However, the statistical efficiency of online learning with respect to KL-regularized objectives remains far from completely characterized, even when specialized to multi-armed bandits (MABs). We address this problem for MABs via a sharp analysis of KL-UCB (Zhao et al., 2025b) using a novel peeling argument, which yields a $\widetilde{O}(\eta K \log^2 T)$ KL-regularized regret upper bound: the *first* high-probability regret bound with linear dependence on $K$. Here, $T$ is the time horizon, $K$ is the number of arms, $\eta^{-1}$ is the regularization intensity, and $\widetilde{O}$ hides all logarithmic factors except those involving $\log T$. The near-tightness of our analysis is certified by the *first* non-constant lower bound $\Omega(\eta K \log T)$, which follows from subtle hard-instance constructions and a tailored decomposition of the Bayes prior. Moreover, in the low-regularization regime (i.e., *large* $\eta$), we show that the KL-regularized regret for MABs is $\eta$-independent and scales as $\widetilde{\Theta}(\sqrt{KT})$. Overall, our results provide a thorough understanding of KL-regularized MABs across all regimes of $\eta$ and yield nearly optimal bounds in terms of $K$, $\eta$, and $T$.

## 1 Introduction

Recently, many variants of the *KL-regularized objective* $J(\pi) := \mathbb{E}_\pi r - \eta^{-1}\mathsf{KL}\left(\pi\|\pi^{\mathsf{ref}}\right)$ have become increasingly important in practice for bandits (Rafailov et al., 2023; Guo et al., 2025) and reinforcement learning (RL) (Schulman

et al., 2017; Ouyang et al., 2022), where $r$ is the mean reward function, $\pi^{\mathsf{ref}}$ is the reference policy, $\eta^{-1}$ is the regularization intensity, and KL is the reverse Kullback-Leibler divergence. For example, they have been instantiated as entropy regularization to strengthen the policy robustness (Williams, 1992; Ziebart et al., 2008; Levine & Koltun, 2013; Haarnoja et al., 2018), and are widely employed in recommender systems (Steck, 2018; Geyik et al., 2019) and large language model fine-tuning (Ouyang et al., 2022; Rafailov et al., 2023; Richemond et al., 2024; Liu et al., 2024; Guo et al., 2025).

Given the prevalence of KL-regularized objectives, a growing body of work has been devoted to understanding the KL-regularized *statistical* efficiency of decision making, where suboptimality is defined with respect to the regularized objective. Xiong et al. (2024); Xie et al. (2025) demonstrate the rate of $\widetilde{O}(\epsilon^{-2})$ for learning an $\epsilon$-optimal policy in contextual bandits and Markov decision processes. Starting from the pioneering Tiapkin et al. (2023); Zhao et al. (2025a), previous works on this line (ignoring other factors) either achieve an $\widetilde{\Theta}(\epsilon^{-1})$ sample complexity (Zhao et al., 2025a; 2026; Foster et al., 2025) or $\mathrm{polylog}(T)$ regret (Zhao et al., 2025b; Wu et al., 2025a) in various interaction protocols. In particular, Tiapkin et al. (2023) obtained a fast-rate sample complexity in the pure exploration setting for both tabular and linear MDPs. Zhao et al. (2025a) works in the hybrid offline setting under a strict uniform data coverage assumption. For online learning, Zhao et al. (2025a) gives the first $\Omega(\eta \log(N_\mathcal{R}))$ regret lower bound[1] that does not scale with the time horizon $T$, and Zhao et al. (2025b) achieves the first logarithmic regret upper bound $\widetilde{O}(\eta d_\mathcal{R} \log(N_\mathcal{R}) \log T)$ under general function approximation, where $d_\mathcal{R}$ is the eluder dimension and $\log(N_\mathcal{R})$ is the metric entropy of the function class, following which Wu et al. (2025a) design an algorithm free of bonus computation, which enjoys an $\widetilde{O}(\exp(\eta) d_\mathcal{R} \log(N_\mathcal{R}) \log T)$ regret. Therefore, all the previous foundational results leave the following problem open.

> *What is the exact regret of* online *learning with*
> KL-regularized *objectives?*

In this paper, we take the first step towards settling this

---

[1]Department of Computer Science, University of California, Los Angeles, CA 90095, USA. Correspondence to: Quanquan Gu <qgu@cs.ucla.edu>.

*Proceedings of the 43$^{rd}$ International Conference on Machine Learning*, Seoul, South Korea. PMLR 306, 2026. Copyright 2026 by the author(s).

[1]See Remark 5.4 for a detailed adaptation and discussion.

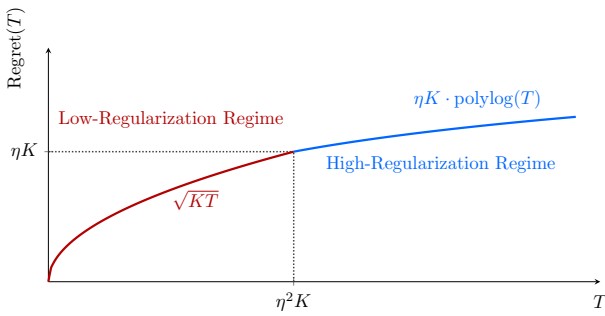

*Figure 1.* The near-comprehensive picture of KL-regularized MABs rendered in this paper. All logarithmic factors except $\log T$ are omitted to avoid clutter.

question via a nearly sharp analysis for multi-armed bandits (MABs), a minimalist model of online learning. In particular, for KL-regularized MABs, we propose a variant of KL-UCB (Zhao et al., 2025b) and provide regret upper bounds in both the high-regularization regime ($\eta$ small) and the low-regularization regime ($\eta$ large). We also construct two sets of hard instances that yield nearly matching regret lower bounds in both regimes, indicating that KL-UCB is near-optimal. Our two-fold contributions are as follows.

- We identify two complementary regimes with different regularization intensities, revealing the transition from $\sqrt{T}$-type regret to $\mathrm{polylog}(T)$-type regret as the regularization strength increases in KL-regularized MABs.

- For the high-regularization regime, our sharp analysis of KL-UCB yields a $\widetilde{O}(\eta K \log^2 T)$ KL-regularized regret. Correspondingly, we also provide a nearly matching $\Omega(\eta K \log T)$ lower bound, characterizing the regret behavior in this regime.

- In the low-regularization regime, our analysis provides an $\widetilde{O}(\sqrt{KT \log T})$ KL-regularized regret upper bound for the same algorithm KL-UCB, which nearly matches our established $\Omega(\sqrt{KT})$ lower bound, similar to the unregularized regret of MABs.

Our near-comprehensive understanding of KL-regularized MABs is visually demonstrated in Figure 1. And relevant bounds on the statistical efficiency of KL-regularized decision making by far are summarized in Table 1 to ease comparison.

**Notation.** The sets $\mathcal{A}$ are assumed to be finite throughout the paper. For nonnegative sequences $\{x_n\}$ and $\{y_n\}$, we write $x_n = O(y_n)$ if $\limsup_{n \to \infty} x_n/y_n < \infty$, $y_n = \Omega(x_n)$ if $x_n = O(y_n)$, and $y_n = \Theta(x_n)$, or alternatively, $y_n \sim x_n$, if $x_n = O(y_n)$ and $x_n = \Omega(y_n)$. We further employ $\widetilde{O}(\cdot), \widetilde{\Omega}(\cdot)$, and $\widetilde{\Theta}(\cdot)$ to hide polylog factors. For

finite $\mathcal{X}$, we denote by $\Delta(\mathcal{X})$ the set of probability distributions on $\mathcal{X}$, and by $\mathsf{Unif}(\mathcal{X})$ the uniform distribution on $\mathcal{X}$. We use $\mathsf{Bern}(p)$ to denote Bernoulli distribution with expectation $p$. For a pair of probability measures $\mathbb{P} \ll \mathbb{Q}$ on the same space, we use $\mathsf{KL}(\mathbb{P}\|\mathbb{Q}) := \int \log(\mathrm{d}\mathbb{P}/\mathrm{d}\mathbb{Q}) \, \mathrm{d}\mathbb{P}$ to denote their KL-divergence. For $p, q \in \mathbb{R}$, we use $\mathcal{N}(p, 1)$ to denote the normal distribution with expectation $p$ and unit variance, and we overload $\mathsf{KL}(p, q)$ to denote the KL-divergence between $\mathcal{N}(p, 1)$ and $\mathcal{N}(q, 1)$. We denote $[N] := \{1, \cdots, N\}$ for any positive integer $N$. Boldfaced lowercase letters, such as $\mathbf{x}$, denote vectors, and their $i$-th entry is denoted by $x_i$. For finite $\mathcal{X}$ and $\mathbf{x}, \mathbf{y} \in \mathcal{X}^n$, we use $d_H(\mathbf{x}, \mathbf{y}) = \sum_{i=1}^n \mathbb{1}(x_i \neq y_i)$ for their Hamming distance. $\forall a, b \in \mathbb{R}$, $a \wedge b := \min\{a, b\}$, $a \vee b := \max\{a, b\}$, and $[a]_{[0,1]} := (a \vee 0) \wedge 1$.

## 2 Related Work

**Optimism in Multi-armed Bandits.** We survey the paradigm of *optimism in the face of uncertainty* for learning finite-armed bandits, which promotes exploration by favoring actions with high uncertainty. The online interactive MAB setting originates from clinical scenarios (Robbins, 1952), where minimizing the *cumulative regret* is vital. Lai & Robbins (1985) initiates the algorithmic principle of optimism for learning MABs and gives the first asymptotic logarithmic regret lower bound. Under the simplification of Bernoulli noise, Lai (1987) proposes the algorithmic paradigm of *upper confidence bound* (UCB), which led to a sequence of works in the asymptotic regime (Agrawal, 1995; Burnetas & Katehakis, 1996). To obtain a finite-time guarantee, Auer et al. (2002a) proposes UCB1, which enjoys finite-time gap-dependent bounds . Audibert & Bubeck (2009) achieves the first worst-case upper bound $O(\sqrt{KT})$ for MABs that is minimax optimal via a UCB-type algorithm (MOSS). This influential UCB paradigm was later extended to be anytime optimal (Degenne & Perchet, 2016) and both minimax and asymptotically optimal (Lattimore, 2018). Beyond classical MABs, the optimism principle has also been adopted in other online decision making problems including bandits with reward function approximation (Abbasi-Yadkori et al., 2011; Chu et al., 2011; Russo & Van Roy, 2013), structured bandits (Kleinberg et al., 2008; Chen et al., 2013), and Markov decision processes (Zhang et al., 2024; Zhou & Gu, 2022).

**RL with KL-Regularization.** Methods that use KL-regularized objectives have achieved strong empirical performance in (inverse) RL and its downstream applications (Ziebart et al., 2008; Schulman et al., 2017; Ouyang et al., 2022; Guo et al., 2025). Several lines of work aim to understand this paradigm. Ahmed et al. (2019); Liu et al. (2021) study the effect of entropy regularization on the stability of policy improvement in policy optimization, and related regret guarantees are analyzed in an online mirror

*Table 1.* Comparison of regret or sample complexity upper and lower bounds for KL-regularized bandits. In this table, $T$ denotes total rounds of interactions, $\epsilon$ the target suboptimality gap, and $\eta$ the KL regularization coefficient. For linear setting, $d$ denotes the dimension of the feature map. For general function approximation, $\mathcal{R}$ is the function class, whose eluder dimension is $d_{\mathcal{R}}$ and covering number is $N_{\mathcal{R}}$, and $C_{\mathrm{GL}}^2$ is an instance-dependent constant that might be arbitrarily large. In the MAB setting, $K$ denotes the number of arm. $\widetilde{O}(\cdot)$ hides logarithmic factors except $\log T$ and $\log(N_{\mathcal{R}})$. A checkmark (✓) indicates that a matching (up to logarithmic factors) lower bound is known for the corresponding setting, while a cross (✗) indicates that no tight lower bound is currently available in its original setting and cannot match the lower bound when specialized to MAB.

| Type | Algorithm | Setting | Regret/Sample Complexity | Matching Lower Bound? |
|---|---|---|---|---|
| Upper Bound | Online Iterative GSHF (Xiong et al., 2024) | Preference w/ Linear Reward | $O(d^2/\epsilon^2)$ | ✗ |
| | TMPS (Zhao et al., 2025a) | Data Coverage | $\widetilde{O}\big((\eta^2 C_{\mathrm{GL}}^2 + \eta/\epsilon)\log(N_{\mathcal{R}})\big)$ | ✗ |
| | Greedy Sampling (Wu et al., 2025a) | Preference w/ Eluder Dimension | $\widetilde{O}\big(\eta \exp(2\eta)d_{\mathcal{R}}\log(N_{\mathcal{R}}T)\big)$ | ✗ |
| | KL-UCB (Zhao et al., 2025b) | Eluder Dimension | $\widetilde{O}\big(\eta d_{\mathcal{R}}\log T\log(N_{\mathcal{R}})\big)$ | ✗ |
| | KL-UCB (This Work) | Multi-armed Bandits | $\widetilde{O}(\eta K \log^2 T)$ | ✓ |
| Lower Bound | Zhao et al. (2025a) | Data Coverage | $\Omega\big(\eta \log(N_{\mathcal{R}})/\epsilon\big)$ | N/A |
| | This Work | Multi-armed Bandits | $\Omega(\eta K \log T)$ | N/A |

descent framework by Cai et al. (2020); He et al. (2022); Ji et al. (2024). Neu et al. (2017) places many KL-regularized algorithms in a unified optimization framework, and subsequent work analyzes the sample complexity of KL/entropy proximal methods in discounted MDPs with improved dependence on the effective horizon (Geist et al., 2019; Vieillard et al., 2020; Kozuno et al., 2022). Nevertheless, because these works measure performance with the unregularized reward objective, the sample complexity for finding an $\epsilon$-optimal policy remains at the statistical lower bound $\Omega(\epsilon^{-2})$.

When switching to performance with respect to the regularized objective, the fast rate $\widetilde{O}(\epsilon^{-1})$ was first established by Tiapkin et al. (2023), who derived a sample complexity of $\widetilde{O}(H^5 S^2 K\eta/\epsilon)$ in the pure exploration setting, where $H$ is the horizon length, $S$ is the cardinality of the context set. Subsequently, Zhao et al. (2025a) obtained a $\widetilde{O}(\eta\epsilon^{-1}\log(N_{\mathcal{R}}))$ sample complexity upper bound, albeit with an additional dependence on a notion of coverage that can be arbitrarily large. Moreover, Zhao et al. (2025a) also provided an $\Omega(\eta \log(N_{\mathcal{R}})\epsilon^{-1})$ sample complexity lower bound, showing that the $\widetilde{O}(\epsilon^{-1})$ rate is optimal. In the regret minimization setting, Zhao et al. (2025b) first obtained an $\widetilde{O}(\eta d_{\mathcal{R}}\log N_{\mathcal{R}}\log T)$ regret upper bound under reward function approximation. Later, Wu et al. (2025a) obtained a $\widetilde{O}(\exp(\eta)d_{\mathcal{R}}\log N_{\mathcal{R}}\log T)$ regret bound without constructing an exploration bonus. This kind of fast convergence against KL-regularized objectives has also been shown for pure offline (Zhao et al., 2026; Foster et al., 2025), competitive multi-agent (Nayak et al., 2025), and privacy-constrained (Wu et al., 2025b; Weng et al., 2025) settings. Nonetheless, no previous results match currently available

worst-case lower bounds with respect to all problem parameters, such as $K$ and $\eta$.

## 3   Problem Setup

We denote a MAB with a KL-regularized objective by a tuple $(\mathcal{A}, r, \eta, \pi^{\mathrm{ref}}, T)$, where $K \coloneqq |\mathcal{A}| < \infty$ is the number of actions, $r : \mathcal{A} \to [0, 1]$ is the reward function unknown to the learner, $\eta > 0$ is the "inverse temperature", $\pi^{\mathrm{ref}} \in \Delta(\mathcal{A})$ is a known reference policy, and $T \geq 1$ is the total number of interactions. At each round $t \in [T]$, the learner selects an action $a_t \in \mathcal{A}$ according to a $\pi_t \in \Delta(\mathcal{A})$ and observes a noisy reward $r_t = r(a_t) + \varepsilon_t$, where $\varepsilon_t$ is 1-sub-Gaussian (Lattimore & Szepesvári, 2020, Definition 5.2). The learner's goal is to minimize the KL-regularized regret:

$$\mathrm{Regret}(T) = \sum_{t=1}^{T}\big[J(\pi^*) - J(\pi_t)\big],$$

where the objective $J(\pi)$ is defined as

$$J(\pi) = \mathbb{E}_{a\sim\pi}\left[r(a) - \eta^{-1}\log\frac{\pi(a)}{\pi^{\mathrm{ref}}(a)}\right]. \quad (3.1)$$

Equivalently, $J(\pi) = \mathbb{E}_{a\sim\pi}[r(a)] - \eta^{-1}\mathsf{KL}(\pi\|\pi^{\mathrm{ref}})$, i.e., the objective subtracts a KL penalty that discourages deviations from the reference policy $\pi^{\mathrm{ref}}$. The regularization strength is controlled by $\eta$: smaller $\eta$ corresponds to stronger regularization.

*Remark* 3.1. For large $\eta$, the effect of KL regularization is small and hence the objective (regret) is close to the objective (regret) without regularization. On the other hand, when $\eta$ is small, the objective forces the learned $\widehat{\pi}$ to stay close to

$\pi^{\mathsf{ref}}$, resulting in a large difference from the unregularized objective and regret. Consequently, an algorithm achieving logarithmic KL-regret might not achieve optimal ordinary cumulative regret.

Under this objective, it is well known that the (unique) optimal policy $\pi^* := \operatorname{argmax}_{\pi \in \Delta(\mathcal{A})} J(\pi)$ has the closed-form expression (see, e.g., Zhang 2023, Proposition 7.16)

$$\pi^*(\cdot) \propto \pi^{\mathsf{ref}}(\cdot) \exp\left(\eta \cdot r(\cdot)\right). \tag{3.2}$$

Moreover, for any reward function $r$, let $\pi_r^*$ denote the corresponding optimal policy. For any policy $\pi \in \Delta(\mathcal{A})$, we define the suboptimality gap of $\pi$ (relative to $\pi_r^*$) by

$$\mathrm{SubOpt}_r(\pi, \pi_r^*) = \mathbb{E}_{a \sim \pi_r^*}\left[r(a) - \eta^{-1}\log\frac{\pi_r^*(a)}{\pi^{\mathsf{ref}}(a)}\right]$$
$$- \mathbb{E}_{a \sim \pi}\left[r(a) - \eta^{-1}\log\frac{\pi(a)}{\pi^{\mathsf{ref}}(a)}\right].$$

*Remark* 3.2. In the case of $\pi^{\mathsf{ref}}(a) = 0$ but $\pi(a) > 0$ for some $a \in \mathcal{A}$, we have $\mathsf{KL}\left(\pi \| \pi^{\mathsf{ref}}\right) = +\infty$ and hence $J(\pi) = -\infty$. This case never arises when $\pi$ is given by $\pi(\cdot) \propto \pi^{\mathsf{ref}}(\cdot) \exp\left(\eta \cdot \widehat{r}(\cdot)\right)$ provided that $r$ is bounded on $\mathcal{A}$.

# 4 Algorithm and Regret Analysis

In this section, we present a variant of KL-UCB (Zhao et al., 2025b), an algorithm for learning MABs with KL-regularization and its corresponding theoretical guarantees.

## 4.1 Algorithm Description

We summarize KL-UCB in Algorithm 1, which follows a similar design to its original version in Zhao et al. (2025b) with general function approximation. In particular, for each round $t \in [T]$, the algorithm first counts the number of times each arm $a$ has been selected, denoted by $N_{t-1}(a)$. Then, the empirical reward is computed using the empirical mean. As in previous works on bandits (Auer et al., 2002a; Zhao et al., 2025b), KL-UCB adopts the principle of optimism in the face of uncertainty (Auer et al., 2002a; Abbasi-Yadkori et al., 2011). Unlike Zhao et al. (2025b), which built the bonus function using the uncertainty with respect to the reward function class, we adopt the following standard bonus for MABs

$$b_t(a) = \sqrt{\frac{2\log(TK/\delta)}{N_t(a) \vee 1}}, \ \forall a \in \mathcal{A}.$$

The following lemma shows that the obtained reward $\widehat{r} = \bar{r} + b$ is indeed an optimistic estimation of the true reward function $r$.

**Lemma 4.1.** *Given $\delta > 0$, let $\mathcal{E}(\delta)$ denote the event that our constructed optimistic reward function is indeed larger than true reward mean, i.e.,*

$$\mathcal{E}(\delta) := \left\{\left|\bar{r}_t(a) - r(a)\right| \le b_t(a), \ \forall(t,a) \in [T] \times \mathcal{A}\right\}.$$

*Then the event $\mathcal{E}(\delta)$ holds with probability at least $1 - \delta$.*

After obtaining the optimistic reward estimation, we construct the policy $\pi_{t+1}$ for time step $t$ to be the optimal policy regarding $\widehat{r}_t$, according to which an action $a_{t+1}$ is sampled and observe the reward $r_{t+1}$.

## 4.2 Theoretical Guarantee

The regret upper bound of Algorithm 1 is given by the following theorem.

**Theorem 4.2.** *With probability at least $1 - 2\delta$, the cumulative regret of Algorithm 1 admits the following upper bounds, depending on the regularization level.*

- *For low regularization ($\eta \ge \sqrt{T/K}$), the regret can be upper bounded as*

$$\mathrm{Regret}(T) = \widetilde{O}\left(\sqrt{KT\log T}\right).$$

- *For high regularization ($\eta \le \sqrt{T/K}$), the regret can be upper bounded as*

$$\mathrm{Regret}(T) = \widetilde{O}\left(\eta K \log^2 T\right),$$

*where $\widetilde{O}$ hides logarithmic factors in $1/\delta$ and $K$.*

*Remark* 4.3. Previously, Zhao et al. (2025b) obtained an $O\left(\eta d(\mathcal{R}, \lambda, T)\log(N_{\mathcal{R}}T/\delta)\right)$ regret under general function approximation, where $\mathcal{R}$ is the reward function class, $d(\mathcal{R}, \lambda, T)$ is the eluder dimension (Zhao et al., 2025b, Definition 3.3) and $N_{\mathcal{R}}$ is the covering number of $\mathcal{R}$. When specializing to MABs, a standard elliptical potential argument (Zhao et al., 2025b, Section 3.1) with one-hot feature mapping shows that $d(\mathcal{R}, \lambda, T) = O(K\log T)$ (Russo & Van Roy, 2013, Section D.1), and $\log N_{\mathcal{R}} = O(K\log T)$. Thus, the worst-case regret upper bound in Zhao et al. (2025b) reduces to $O(\eta K^2 \log^2 T)$ in the multi-armed setting. Compared with their result, the $O(\eta K \log^2 T)$ regret in Theorem 4.2 is strictly better.

Theorem 4.2 establishes the regret upper bound of Algorithm 1 in two separate regimes. When $\eta \ge \sqrt{T/K}$, the regret scales with $\widetilde{O}(\sqrt{KT})$. In contrast, when the regularization is high, i.e., $\eta \le \sqrt{T/K}$, Algorithm 1 enjoys a logarithmic regret $O(\eta K \log^2 T)$. These two regimes arise from the two-term structure of the KL-regularized objective (3.1). When $\eta$ is large, the effect of the regularization term becomes negligible, so the reward term dominates. In this case, the problem resembles a standard MAB problem and therefore recovers the $\widetilde{O}(\sqrt{KT})$ rate. Otherwise, the KL regularization term dominates. It introduces sufficient curvature into the reward estimation error, thereby yielding logarithmic regret.

---

**Algorithm 1** KL-regularized Upper Confidence Bound Algorithm (KL-UCB)

---

**Require:** Regularization $\eta$, reference policy $\pi^{\mathsf{ref}}$, total rounds of interaction $T$, number of actions $K$, error probability $\delta$.
1: **for** $t = 0, ..., T - 1$ **do**
2:      Set $N_t(a) = \sum_{i=1}^{t} \mathbb{1}\{a_i = a\}$ for all $a \in \mathcal{A}$
3:      Compute the empirical reward $\bar{r}_t(a)$ and bonus $b_t(a)$ as

$$\bar{r}_t(a) \leftarrow \frac{1}{N_t(a) \vee 1} \sum_{i=1}^{t} r_i \mathbb{1}\{a_i = a\}, \quad b_t(a) \leftarrow \sqrt{\frac{2 \log(TK/\delta)}{N_t(a) \vee 1}}$$

4:      Set $\widehat{r}_t(a) \leftarrow [\bar{r}_t(a) + b_t(a)]_{[0,1]}$
5:      Compute $\pi_{t+1}(a) \propto \pi^{\mathsf{ref}}(a) \exp\left(\eta \cdot \widehat{r}_t(a)\right)$, play action $a_{t+1} \sim \pi_{t+1}$, and observe $r_{t+1}$
6: **end for**

---

### 4.3 Proof Sketch of Theorem 4.2

The proof operates on the high-probability event $\mathcal{E}(\delta)$, where the optimistic estimator satisfies $|\widehat{r}(a) - r(a)| \leq b_t(a)$. In the low regularization regime, we follow the usual UCB-type analysis routine (Lattimore & Szepesvári, 2020). The intriguing regime of high regularization is analyzed as follows. The full version is deferred to Appendix B.

We begin with the KL-regularized regret decomposition (Lemma B.1). Under $\mathcal{E}_{(\delta)}$, the regret is bounded by the cumulative expected squared error, which scales with the inverse visitation counts:

$$\mathrm{Regret}(T) \leq \eta \sum_{t=1}^{T} \mathbb{E}_{a \sim \pi_t}\left[\left(\widehat{r}(a) - r(a)\right)^2\right]$$
$$\lesssim \eta \sum_{t=1}^{T} \mathbb{E}_{a \sim \pi_t}\left[\frac{1}{N_{t-1}(a) \vee 1}\right]. \quad (4.1)$$

To bound this sum of expectations, we decompose (4.1) into an on-policy term ($I_1$) and a martingale difference term ($I_2$): $(4.1) = \eta(I_1 + I_2)$, (we omit the $\cdot \vee 1$ in this sketch to avoid notation clutter) where

$$I_1 := \sum_{t=1}^{T} 1/N_{t-1}(a_t),$$
$$I_2 := \sum_{t=1}^{T} \left(\mathbb{E}_{a \sim \pi_t}[1/N_{t-1}(a)] - 1/N_{t-1}(a_t)\right).$$

**Bounding $I_1$ (Harmonic Sum).** The realized term is bounded deterministically via the properties of the harmonic series. A double counting with respect to arms yields

$$I_1 = \sum_{a \in \mathcal{A}} \sum_{i=1}^{N_{T-1}(a)} \frac{1}{i} \approx \sum_{a \in \mathcal{A}} \log\left(N_{T-1}(a)\right) \lesssim K \log T.$$

**Bounding $I_2$ (Peeling Technique).** The terms in $I_2$ form a martingale difference sequence (MDS). While it can be safely ignored if we only require a guarantee on the expected regret, obtaining a high-probability regret upper bound requires a concentration argument. A straightforward invocation of the Azuma-Hoeffding inequality yields an $\widetilde{O}(\sqrt{T})$ bound. Although such a bound is acceptable in the unregularized setting, where the overall regret is also of order $\widetilde{O}(\sqrt{T})$, it would dominate and thus destroy the desired $O(\log T)$ bound in the KL-regularized setting. Therefore, we need a more delicate approach.

In particular, let $x_t$ be an element of this MDS:

$$x_t = \mathbb{E}_{a \sim \pi_t}\left[\frac{1}{N_{t-1}(a)}\right] - \frac{1}{N_{t-1}(a_t)}.$$

and let $y_t := 1/N_{t-1}(a_t)$ be the on-policy term. Since $\sum_t y_t \lesssim K \log T$, the typical magnitude of $x_t$ is much smaller than its trivial upper bound 1, suggesting that the cumulative conditional variance of $\{x_t\}$ should also be small. This observation motivates the application of Freedman's inequality (Lemma D.2) for bounding $I_2 = \sum_t x_t$ via the sum of conditional variances. However, a direct application requires a sufficiently tight bound on the conditional variance; using only a crude upper bound would yield $O(\sqrt{T})$ regret bound, and thus fail to obtain the desired fast rate. To resolve this issue, we employ a novel *peeling technique*. Specifically, we use the inequality $\mathrm{Var}(x_t | \mathcal{F}_{t-1}) \leq \mathbb{E}[y_t | \mathcal{F}_{t-1}]$ and truncate $\sum_t \mathbb{E}[y_t | \mathcal{F}_{t-1}]$ at different levels $2^i$ for a more fine-grained upper bound of $\sum_t \mathrm{Var}(x_t | \mathcal{F}_{t-1})$. For each $i > 0$, we define the following event

$$\mathcal{E}_i(t) = \left\{\sum_{s=1}^{t} \mathbb{E}[y_s | \mathcal{F}_{s-1}] \leq 2^i\right\}.$$

Since $\sum_{t=1}^{T} \mathbb{E}[y_t | \mathcal{F}_{t-1}] \leq T$ surely, we only need to consider $i \leq \log_2 T$. It is easy to check that for any $i$, $\{x_t \mathbb{1}(\mathcal{E}_i(t))\}_{t=1}^{T}$ remains a MDS. Moreover, its conditional variance can be upper bounded by $2^i$, i.e.,

$$\sum_{t=1}^{T} \mathrm{Var}\left(x_t \mathbb{1}[\mathcal{E}_i(t)] \big| \mathcal{F}_{t-1}\right) \leq 2^i.$$

Now we apply Freedman's inequality to $\{x_t \mathbb{1}(\mathcal{E}_i(t))\}_{t=1}^T$. Taking a union bound over all $i \le \log_2 T$, we conclude that with high probability, the following inequality holds simultaneously for all $i \le \log_2 T$,

$$\sum_{t=1}^T x_t \mathbb{1}[\mathcal{E}_i(t)] \le \widetilde{O}(2^{i/2}) + \text{minor terms.} \quad (4.2)$$

We now select an index $i$ such that $\mathbb{1}[\mathcal{E}_i(t)] = 1, \forall t$. Since $\sum_{t=1}^T \mathbb{E}[y_t | \mathcal{F}_{t-1}] \le T$ and (4.2) holds for all $i \le \log_2 T$ simultaneously, we can choose $i$ such that $2^i \sim \sum_{t=1}^T \mathbb{E}[y_t | \mathcal{F}_{t-1}]$. Then, (4.2) gives

$$I_2 \lesssim \sqrt{\sum_{t=1}^T \mathbb{E}_{a \sim \pi_t}\left[\frac{1}{N_{t-1}(a)}\right]} + \text{minor terms.}$$

Finally, recalling that $I_1 + I_2 = \sum_{t=1}^T \mathbb{E}_{a \sim \pi_t}[1/N_{t-1}(a)]$ and $I_1 \lesssim K \log T$, we have

$$(I_1 + I_2) \lesssim K \log T + \sqrt{I_1 + I_2} + \text{minor terms.}$$

We can conclude the proof using the resulting quadratic inequality $X \le A\sqrt{X} + B \implies X \lesssim A^2 + B$.

## 5 Lower Bounds

In this section, we present two minimax lower bounds to show that KL-UCB is nearly minimax optimal. We first present the lower bound in the low-regularization regime, where $\eta \ge \sqrt{T/K}$.

**Theorem 5.1** (Low-regularization regime). *Given any $K \ge 9$ and $\eta \ge \sqrt{T \log^2 K / K}$, for any algorithm, there exists a KL-regularized $K$-armed bandit on which the algorithm suffers from $\Omega(\sqrt{KT})$ regret.*

A change-of-variable argument $\widetilde{T} \leftarrow T/\log^2 K$ then yields the following corollary in the regime of $\eta \ge \sqrt{T/K}$.

**Corollary 5.2.** *Given any $K \ge 9$ and $\eta \ge \sqrt{T/K}$, for any algorithm, there exists a KL-regularized $K$-armed bandit on which the algorithm suffers from $\Omega(\sqrt{KT} \log^{-1} K)$ regret.*

In the high-regularization regime where $\eta \le \sqrt{T/K}$, the regret lower bound is characterized by the following theorem.

**Theorem 5.3** (High-regularization regime). *Given any $K \ge 2$, $0 < \eta \le \sqrt{T/K}$, for any algorithm, there exists a KL-regularized $K$-armed bandit on which the algorithm suffers from $\Omega\big(\eta K \log\big(T/(\eta^2 K)\big)\big)$ regret.*

*Remark* 5.4. Previously, an $\Omega(\eta \log N_\mathcal{R}/\epsilon)$ lower bound was introduced in Zhao et al. (2025a) for a 2-armed contextual bandit, which implies an $\Omega(\eta/\epsilon)$ sample complexity for MABs[2]. In contrast, Theorem 5.3 establishes an

$\Omega(\eta K \log T)$ lower bound and implies[3] an $\Omega(\eta K/\epsilon)$ sample complexity, strictly improving upon the previous result.

When $\eta \ge \sqrt{T/K}$, Theorem 5.1 shows that any algorithm must incur $\widetilde{\Omega}(\sqrt{KT})$ regret. On the other hand, when $\eta \le \sqrt{T/K}$, Theorem 5.3 shows that any algorithm must incur $\Omega(\eta K \log T)$ regret. Compared with the upper bound in Theorem 4.2, these lower bounds together show that KL-UCB is *minimax optimal* up to logarithmic factors, and the logarithmic dependence on $T$ in the high-regularization regime is *inevitable*.

Remarkably, a $\log T$ gap remains between the upper bound in Theorem 4.2 and the lower bound in Theorem 5.3. We conjecture that this gap primarily arises from the high-probability nature of Theorem 4.2: the analysis requires a union bound over all $T$ rounds, which introduces an additional $\log T$ factor. Closing this gap, potentially through an expected-regret analysis of minimax-optimal bandit algorithms such as MOSS (Audibert & Bubeck, 2009) in the KL-regularized setting, remains an interesting direction for future work.

## 6 Proof Overview of Hardness Results

In this section, we provide an overview of the proofs in Section 5. We first discuss the proof of Theorem 5.1, which corresponds to the low-regularization regime and more similar to unregularized MABs. Accordingly, following previous works (Lattimore & Szepesvári, 2020), we construct a hard instance class consisting of $K$ hard-to-distinguish instances. However, this is not sufficient for proving the lower bound in the high-regularization regime (Theorem 5.3). We first explain why the classical construction fails, and then propose a new approach based on a more sophisticated family of instances. Throughout this section, we assume that the reward noise is independent Gaussian with variance 1 unless otherwise specified.

### 6.1 Proof Overview of Theorem 5.1

In this theorem, we consider the low-regularization regime, where $\eta \gtrsim \sqrt{T \log^2 K / K}$. In this regime, the effect of regularization is negligible, thus the regularized problem can be viewed as a small perturbation of the unregularized bandit. Accordingly, we construct hard instances by adapting the standard unregularized bandit lower-bound construction (see, e.g., Lattimore & Szepesvári (2020)). Specifically, we fix $\eta > 0$, $\mathcal{A} = [K]$, $\pi^{\text{ref}} = \text{Unif}(\mathcal{A})$. We construct the hard-to-distinguish instance set as follows:

Fix a constant $\delta > 0$ to be specified later. For the

---

[2]The $\log N_\mathcal{R}$ scaling entirely arises from the size of the context set and hence reduces to a constant in MABs.

[3]While in general a regret lower bound does not imply a sample complexity lower bound, the sample complexity lower bound here can be derived from the proof of Theorem 5.3. We refer to Remark C.1 for details.

first instance, we define the reward function $r_1$ by setting $r_1(1) = \delta$ and $r_1(i) = 0$ for all $i \geq 2$. For the remaining instances, for each $k \in 2, \ldots, K$, we define $r_k$ by setting $r_k(i) = r_1(i)$ for all $i \neq k$ and $r_k(k) = 2\delta$.

For any algorithm Alg, let $N_T(i)$ be the number of times arm $i$ is pulled in the first $T$ steps. By the pigeonhole principle, there exists $k \geq 2$ such that

$$\mathbb{E}_{r_1, \mathsf{Alg}}[N_T(k)] \leq \frac{T}{K-1},$$

where the expectation is taken over the distribution jointly given by instance $r_1$ and Alg. Now we consider the KL-divergence between the trajectory distributions induced by instances $r_1$ and $r_k$. By the chain rule of KL-divergence and $\mathsf{KL}(0, 2\delta) = 2\delta^2$ (Lemma D.1), $\mathsf{KL}(\mathbb{P}_1 \| \mathbb{P}_k)$ can be bounded by

$$\sum_{i=1}^{K} \mathbb{E}_1[N_T(i)] \mathsf{KL}(r_1(i), r_k(i)) \leq \frac{T\delta^2}{K-1}, \qquad (6.1)$$

where we adopt the shorthand $\mathbb{P}_i := \mathbb{P}_{r_i, \mathsf{Alg}}$ to denote the probability distributions over trajectories induced by the interaction between algorithm Alg and instances $r_i$ for $i \in [K]$. When picking $\delta \sim \sqrt{K/T}$, we have $\mathsf{KL}(\mathbb{P}_1 \| \mathbb{P}_k) = O(1)$, indicating that under algorithm Alg, it is hard to distinguish $r_k$ from $r_1$[4].

Now we compute the cost of misidentifying the underlying reward function. For $i \in \{1, k\}$, let $\pi_i^*$ be the optimal policy corresponding to $r_i$, as defined in (3.2). We define the suboptimality gap between $\pi_i^*$ and any policy $\pi \in \Delta(\mathcal{A})$ under instance $i$ as $\mathrm{SubOpt}_i(\pi) = \mathrm{SubOpt}_{r_i}(\pi, \pi_i^*)$. A direct computation yields that

$$\mathrm{SubOpt}_1(\pi) + \mathrm{SubOpt}_k(\pi)$$
$$\gtrsim \frac{1}{\eta} \log \frac{(e^{\eta\delta} + K - 1)(e^{2\eta\delta} + K - 2)}{(2e^{\eta\delta} + K - 2)^2}. \qquad (6.2)$$

As $\delta \sim \sqrt{K/T}$, our assumption on $\eta$ indicates $\eta\delta \gtrsim \log K$. Then, all the $\Theta(K)$ term in the denominator of (6.2) can be replaced by $\exp(\eta\delta)$, which yields

$$\mathrm{SubOpt}_1(\pi) + \mathrm{SubOpt}_k(\pi) \gtrsim \frac{1}{\eta} \log \frac{e^{\eta\delta} \cdot e^{2\eta\delta}}{(3e^{\eta\delta})^2}$$
$$\gtrsim \delta. \qquad (6.3)$$

Consequently, (6.3) shows that the algorithm incurs a per-step cost of $\Omega(\delta)$ if it cannot distinguish $r_k$ from $r_1$ and therefore suffers $\Omega(T\delta)$ regret over $T$ rounds. Now we combine (6.1) and (6.3) with an argument of Le Cam's method (Lemma D.5), we conclude that Alg suffers from $\Omega(T\delta) = \Omega(\sqrt{KT})$ regret, which finishes the proof.

---

[4]In general, the KL-divergence $\mathsf{KL}(\mathbb{P} \| \mathbb{Q})$ between two distributions $\mathbb{P}$ and $\mathbb{Q}$ is a constant indicates that $\mathbb{P}$ and $\mathbb{Q}$ cannot be reliably distinguished.

## 6.2 Proof Overview of Theorem 5.3

Although the hard instance constructed in Section 6.1 is standard for MABs, it does not apply to the high-regularization (fast-rate) case. In this section, we first explain why this construction fails in that regime. To overcome the difficulty, we then introduce new proof techniques to derive a sharper fast-rate lower bound.

**Failure of Instances in Section 6.1.** At a high level, the lower bound proof in Section 6.1 relies on two key steps: constructing a set of statistically indistinguishable instances by setting $\delta = \sqrt{K/T}$ (6.1), and demonstrating that the suboptimality is sufficiently large on at least one of these instances (6.3).

We first demonstrate that, in the regime of high regularization, the curvature of the regularizer plays a crucial role, resulting to a $\Omega(\eta\delta^2)$ rather than $\Omega(\delta)$ suboptimality gap. Specifically, when $\eta$ is small, we can redo (6.2) as follows:

$$(6.2) = \frac{1}{\eta} \log \left( 1 + \frac{M}{(M + \exp(\eta\delta))^2} \left( \exp(\eta\delta) - 1 \right)^2 \right),$$

where $M = K - 2 + \exp(\eta\delta)$. When $\eta\delta$ is small, $K - 2$ dominates $\exp(\eta\delta)$ and make $M = \Omega(K)$. Now, applying basic inequality regarding $\eta\delta$ results in

$$(6.2) \sim \frac{1}{\eta} \log \left( 1 + \frac{\eta^2 \delta^2}{K} \right) \sim \frac{\eta\delta^2}{K}. \qquad (6.4)$$

Ignoring the dependence on $K$, we see that the suboptimality gap is of order $\eta\delta^2$. Moreover, using the choice of $\delta \sim \sqrt{K/T}$, we obtain an $\Omega(\eta)$ regret bound, whose dependency on $K$ is loose compared with Theorem 5.3.

The gap with respect to $K$ is primarily due to the fact that strong regularization toward $\pi^{\mathsf{ref}} = \mathsf{Unif}(\mathcal{A})$ forces any near-optimal policy to remain close to the uniform policy. Consequently, the policy assigns only $O(1/K)$ probability mass to the specific arms $\{1, k\}$ where the instances differ. As a result, the cost of making an error in distinguishing $r_k$ from $r_1$ is diluted by a factor of $K$, i.e., from $\Omega(\eta\delta^2)$ to $\Omega(\eta\delta^2/K)$. Therefore, to manifest the $\Omega(K)$ dependency in Theorem 5.3, we need a more sophisticated set of instances.

*Remark* 6.1. Since the two-point-type constructions in the proofs of lower bounds in previous works on KL-regularized decision making (Zhao et al., 2025a; 2026) are in spirit similar to the construction for Theorem 5.1 *when specialized to MABs*, the reasoning above also implies that it is not promising to directly adapt their constructions to the online setting to show the correct scaling with respect to $K$.

**Instance Design.** To overcome the issue above, we instead consider a class of instances in which $\Omega(K)$ arms might have different rewards and thus require estimation. In particular, let $K$ be even and $A := K/2$. We fix $\eta > 0$ and

keep $\mathcal{A} = [K]$ and $\pi^{\text{ref}} = \text{Unif}(\mathcal{A})$. Let $\mathcal{V} = \{\pm 1\}^A$ and we consider the rewards parameterized by $\boldsymbol{\mu} \in \mathcal{V}$ such that $r_{\boldsymbol{\mu}}$ is given as follows:

$$r_{\boldsymbol{\mu}}(i) = \frac{1}{2} + \boldsymbol{\mu}_i \delta, \ \forall i \in [A];$$

$$r_{\boldsymbol{\mu}}(i) = \frac{1}{2}, \ \forall i \in [A+1, 2A],$$

where $\delta > 0$ is a parameter to be specified. Upon this set of instances, to distinguish one of the reward $r_{\boldsymbol{\mu}}$ from all other rewards in $\{r_{\boldsymbol{\nu}}\}_{\boldsymbol{\nu} \in \mathcal{V}}$, the learner has to determine all the $A = \Omega(K)$ entries of $\boldsymbol{\mu}$.

**Suboptimality Gap Computation.** Our next step is to demonstrate that the regret accumulates across all arms where the learner fails to distinguish whether the mean reward is $1/2 + \delta$ or $1/2 - \delta$. Intuitively, for any $\boldsymbol{\mu} \in \mathcal{V}$ and $i \in [K]$, $r_{\boldsymbol{\mu}}(i)$ is very close to $1/2$ and therefore all near-optimal policies put $\Theta(1/K)$ probability mass on each arm. Hence, similar to the argument of (6.4), once the learner makes an error in estimating some $r(k)$, the cost of this error is always $\Omega(\eta\delta^2/K)$ regardless of the estimation on the other arms. Therefore, the cost accumulates and results in $\Omega(m\eta\delta^2/K)$ total cost if learner makes $m$ mistakes.

In particular, let $\boldsymbol{\mu}_1, \boldsymbol{\mu}_2 \in \mathcal{V}$ be two instances such that $d_H(\boldsymbol{\mu}_1, \boldsymbol{\mu}_2) = m$. From now on, for $i = 1, 2$, let $r_i = r_{\boldsymbol{\mu}_i}$, $\pi_i^*$ be the optimal policy corresponding to $r_i$ and $\text{SubOpt}_i(\pi) = \text{SubOpt}_{r_i}(\pi, \pi_i^*)$ be the suboptimality gap between $\pi_i^*$ and any policy $\pi \in \Delta(\mathcal{A})$. A direct computation yields that

$$\text{SubOpt}_1(\pi) + \text{SubOpt}_2(\pi)$$
$$\gtrsim \frac{1}{\eta} \log \left( 1 + \frac{2Km}{\left( K \exp(\eta\delta) \right)^2} \left( e^{\eta\delta/2} - 1 \right)^2 \right). \quad (6.5)$$

In promise that $\eta\delta = O(1)$, $\exp(\eta\delta) = O(1)$ and the $\exp(\eta\delta)$ in the denominator in (6.5) can be ignored and then (6.5) becomes

$$\text{SubOpt}_1(\pi) + \text{SubOpt}_2(\pi)$$
$$\gtrsim \frac{1}{\eta} \log \left( 1 + \frac{m}{K} \left( \exp(\eta\delta/2) - 1 \right)^2 \right)$$
$$\gtrsim \frac{1}{\eta} \log \left( 1 + \frac{m}{K} \eta^2 \delta^2 \right) \gtrsim \frac{m\eta\delta^2}{K}, \quad (6.6)$$

where the second inequality holds due to $e^x - 1 \approx x$ and the last holds due to $\log(1+x) \approx x$. Consequently, (6.6) demonstrates that if the algorithm fails to distinguish between instances with $m$ arms differ, it suffers a per-step cost of $\Omega(m\eta\delta^2/K)$.

**Minimax Lower Bound of the Suboptimality Gap.** We show that for $t \geq \eta^2 K$, there exists a choice of $\delta_t$ such that

the suboptimality gap at time step $t$ is $\Omega(\eta K/t)$. Fixing $t \geq \eta^2 K$, we pick $\delta_t = \sqrt{K/t}$, and use $\boldsymbol{\mu} \sim_j \boldsymbol{\lambda}$ to denote $d_H(\boldsymbol{\mu}, \boldsymbol{\lambda}) = 1$ and $\boldsymbol{\mu}_j \neq \boldsymbol{\lambda}_j$. As in (6.1), we consider the average KL-divergence (up to round $t$) between pairs of instances which differ only on arm $j$:

$$\frac{1}{|\mathcal{V}|} \sum_{\boldsymbol{\mu} \sim_j \boldsymbol{\lambda}} \text{KL}(\mathbb{P}_{\boldsymbol{\mu},t} \| \mathbb{P}_{\boldsymbol{\lambda},t}).$$

Averaging over $j \in [A]$, one can show that

$$\frac{1}{A} \sum_{j=1}^{A} \frac{1}{|\mathcal{V}|} \sum_{\boldsymbol{\mu} \sim_j \boldsymbol{\lambda}} \text{KL}(\mathbb{P}_{\boldsymbol{\mu},t} \| \mathbb{P}_{\boldsymbol{\lambda},t}) = \frac{2t\delta_t^2}{K} = 2.$$

Consequently, there exist $m = \Omega(K)$ arms for which the corresponding average KL-divergences are $O(1)$, implying that the algorithm Alg cannot reliably distinguish the rewards on these arms. Plugging $m = \Omega(K)$ into (6.6) gives $\Omega(\eta K/t)$ suboptimality gap.

**Summing Over Time Steps.** If we ignore small time steps, directly summing the $\Omega(\eta K/t)$ suboptimality gap for every $t \in [\lceil \eta^2 K \rceil, T]$ up yields an expected regret lower bound of $\Omega(\eta K \log(T/(\eta^2 K)))$. Such an approach is, however, flawed since the $\delta_t$ is different for each $t$. This temporal-level discrepancy of the set of hard instances prevents a direct aggregation of these bounds, because the trajectory distribution would be ill-defined if the instances keep evolving as $t$ grows from $1$ to $T$.

To overcome this issue, we construct a single collection of instances that remains invariant for all $t = \Omega(\eta^2 K)$. The idea here is to extend the discrete instance distribution to a continuous distribution. Similar proof ideas have also been applied in previous works (Vovk, 2001; Singer et al., 2002; Zhao et al., 2023) to derive $\log T$-type lower bounds. For clarity, we illustrate the idea under $K = 2$, in which $\mathcal{V} = \{\pm 1\}$.

Fixing some $t$ and the corresponding reward gap $\delta$, the rewards in the previous construction are distributed over $1/2 \pm \delta$. To make this distribution continuous, we replace $1/2$ with a variable $x$ ranging from $1/2 - \delta$ to $1/2 + \delta$. Then $x - \delta$ exactly scans over $[1/2 - 2\delta, 1/2]$ and $x + \delta$ exactly scans over $[1/2, 1/2 + 2\delta]$, which, collectively, constitutes a uniform coverage of a $4\delta$-length interval. Moreover, pairing the rewards as $x \pm \delta$ and applying (6.6) preserves the lower bound:

$$\mathbb{E}_{x \sim \text{Unif}([1/2-\delta, 1/2+\delta])} \mathbb{E}_{v \in \mathcal{V}} \mathbb{E}_{r_{x+v\delta,t}} [\text{SubOpt}_{r_{x+v\delta,t}}(\pi)]$$
$$= \mathbb{E}_{u \sim \text{Unif}([1/2-2\delta, 1/2+2\delta])} \mathbb{E}_{r_u,t} [\text{SubOpt}_{r_u,t}(\pi)]$$
$$\gtrsim \eta\delta^2.$$

We then concatenate several consecutive and disjoint copies of the $4\delta$-length interval to form an interval of length $\alpha$. A

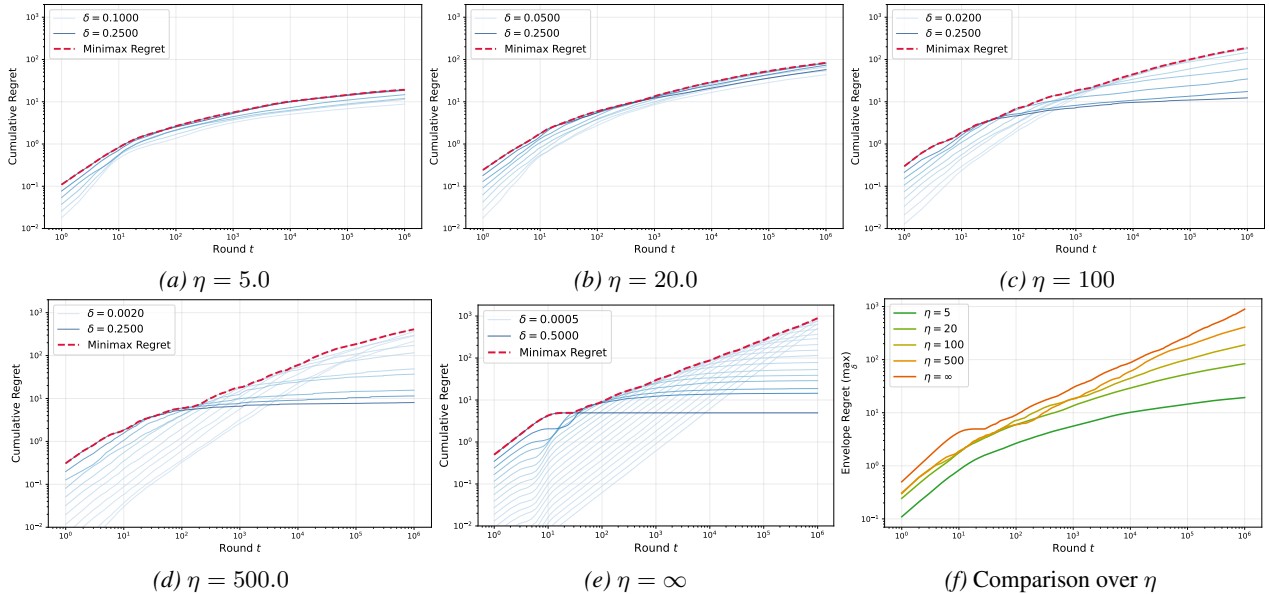

*Figure 2.* The regret curves of KL-UCB for different values of $\eta$, with both axes displayed on log scales. For each fixed $\eta$, the regret corresponding to each instance (determined by $\delta$) is plotted as a single blue solid line in Figures 2a–2e. The red bold dashed lines represent the maximum regret over all instances, and thus approximate the minimax regrets. For ease of comparison, all minimax regret curves are plotted together in Figure 2f.

uniform distribution of instances over the $\alpha$-length interval still yields an $\Omega(\eta\delta)$ suboptimality gap lower bound.

Now, for each $t \geq \eta^2 K$, we first pick $\delta_t = \sqrt{K/t}$, and then, if $\alpha$ is sufficiently large, a slight adjustment to $\delta_t$ enables $\alpha/(2\delta_t) \in \mathbb{N}^*$ so that the construction above produces an $t$-independent uniform distribution over an interval of length $\alpha$. This addresses the issue of varying instance distributions across time $t$. Now we can sum over all $t = \Omega(\eta^2 K)$ and obtain the $\Omega(\eta K \log T)$ regret lower bound as desired.

## 7 Experiments

In this section, we conduct experiments to verify the minimax regret guarantees and phase transition presented in Section 4 and Section 5. we also investigate the policy convergence of Algorithm 1 and defer the details to Appendix A.

In this experiment, we run Algorithm 1 on a set of KL-regularized $K$-armed bandits and fix $K = 4$, $T = 10^6$, $\pi^{\mathsf{ref}} = \mathsf{Unif}(\mathcal{A})$ and rewards $r = [\delta, 2\delta, 0, 0]$. To demonstrate how the strength of regularization affects the regret behavior, we consider $\eta \in \{5, 20, 100, 500, \infty\}$, where $\eta = \infty$ corresponds to the standard MAB setting. For each fixed $\eta$, we sweep $\delta$ log-uniformly over an appropriate interval, and plot the regret curve of each instance (i.e., a choice of $\delta$). We further take the maximum regret over all instances to approximate the minimax regret. The regret curves are compiled in Figure 2. We observe that the minimax regret curve for the standard MAB setting exhibits a

slope of $1/2$ for $t > 100$, indicating a $\sqrt{T}$ rate. In contrast, when $\eta < \infty$, the slope of the regret curve decreases gradually as $t$ increases, with more pronounced decreases for smaller values of $\eta$. These results are consistent with the theoretical guarantees established in the paper.

## 8 Conclusion and Future Work

In this work, we study the MAB problem with a KL-regularized objective and provide a near-complete characterization of the behavior of KL-regularized regret. In particular, we propose a variant of KL-UCB (Zhao et al., 2025b) that achieves a $\widetilde{O}(\eta K \log^2 T)$ regret upper bound. This regret is near-optimal, as indicated by an $\Omega(\eta K \log T)$ regret lower bound. Furthermore, in the low regularization regime, our theoretical analysis shows an $\widetilde{\Theta}(\sqrt{KT \log T})$ regret with matching bounds, providing a comprehensive understanding of the KL-regularized objectives for *online* learning in MABs.

Currently, there is still a $\Theta(\log T)$ gap between our upper and lower bounds. Moreover, our analysis is restricted to the tabular setting with finitely many arms and stochastic rewards. Fully closing the gap and extending these near-matching results to structured settings such as contextual bandits (Chu et al., 2011), bandits with linear or general function approximation (Abbasi-Yadkori et al., 2011; Russo & Van Roy, 2013) and decision making in the face of adversary (Auer et al., 2002b) are interesting directions for future work.

## Acknowledgements

We thank the anonymous reviewers and area chair for their helpful comments. We also thank Weitong Zhang and Xuheng Li for helpful discussions at the early stage of this project. KJ, QZ, HZ, QD and QG are supported in part by the National Science Foundation DMS-2323113, IIS-2403400 and CPS-2312094. The views and conclusions contained in this paper are those of the authors and should not be interpreted as representing any funding agencies.

## Impact Statement

This paper presents work whose goal is to advance the field of Online Learning. There are many potential societal consequences of our work, none which we feel must be specifically highlighted here.

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

# A    Additional Experiments

In this section, we present some empirical policy convergence results of Algorithm 1. In these experiments, we consider the $K$-armed bandits with $K = 5$, $T = 10^6$, $\pi^{\mathsf{ref}} = \mathsf{Unif}(\mathcal{A})$ and reward randomly sampled from $\mathsf{Unif}([0,1]^5)$. To characterize the effect of the regularization strength, we vary $\eta \in \{2, 10, 50\}$. For each $\eta$, we plot the TV distance between the learned policy $\pi_t$ and three comparison policies: the reference policy $\pi_{\mathrm{ref}}$, the optimal policy $\pi^*_{\mathrm{KL}}$ for the KL-regularized regret, and the deterministic optimal policy $\pi^*_{\mathrm{std}}$ for the standard regret. As shown in Figure 3, for small $\eta$ (strong regularization), $\pi_t$ remains close to $\pi_{\mathrm{ref}}$ and converges toward $\pi^*_{\mathrm{KL}}$, consistent with the theoretical prediction that the KL penalty dominates the objective. As $\eta$ increases, the regularization weakens and the policy drifts away from $\pi_{\mathrm{ref}}$, instead approaching $\pi^*_{\mathrm{std}}$, thereby recovering the unregularized optimum. This behavior illustrates how $\eta$ interpolates between the two regimes, complementing our theoretical guarantees on policy convergence.

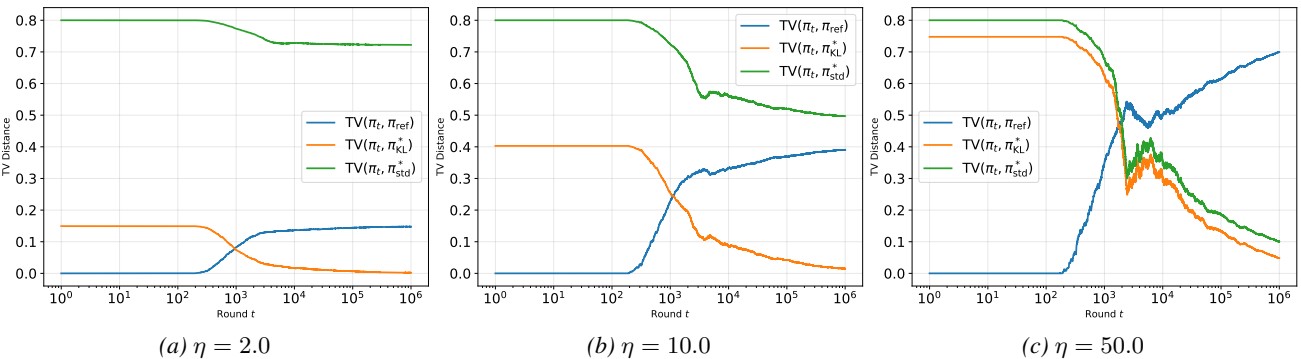

*(a)* $\eta = 2.0$  *(b)* $\eta = 10.0$  *(c)* $\eta = 50.0$

*Figure 3.* Policy convergence in TV distance, with $\eta \in \{2, 10, 50\}$. As $\eta$ increases, the regularization weakens, making the policy drifts away from the reference policy $\pi_{\mathrm{ref}}$ and approaches the optimal policy under standard un-regularized objective $\pi^*_{\mathrm{std}}$.

# B    Missing Proof in Section 4

## B.1    Proof of Lemma 4.1

*Proof of Lemma 4.1.* The proof is standard and we present here for completeness. Fix a time step $t$ and a specific arm $a$, by Hoeffding's inequality (Lemma D.3), with probability at least $1 - \delta/KT$, we know that

$$r(a) - \frac{1}{N_t(a) \vee 1} \sum_{i=1}^{t} r_i \, \mathbb{1}\{a_i = a\} \le \sqrt{\frac{2\log(KT/\delta)}{N_t(a) \vee 1}} = b_t(a).$$

Taking union bound over all $t \in \overline{[T]}$ and $a \in \mathcal{A}$ finishes the proof.[5]  $\qquad\square$

## B.2    Proof of Theorem 4.2

*Proof of Theorem 4.2.* The proof follows the the previous proof in Zhao et al. (2025b). We first prove the "fast rate" when $\eta$ is small. The following lemma gives the KL-regularized regret decomposition.

**Lemma B.1** (Lemma A.1, Zhao et al. 2025b)**.** *Let $\widehat{r}$ be an optimistic estimator of the ground truth reward $r$, i.e., $\widehat{r}(a) \ge r(a)$ for all $a \in \mathcal{A}$. Let $\widehat{\pi}(a) \propto \pi^{\mathsf{ref}}(a) \exp\big(\eta \cdot \widehat{r}(a)\big)$ and $\pi^*(a) \propto \pi^{\mathsf{ref}}(a) \exp\big(\eta \cdot r(a)\big)$, then*

$$J(\pi^*) - J(\widehat{\pi}) \le \eta \mathbb{E}_{a \sim \widehat{\pi}}\big[\big(\widehat{r}(a) - r(a)\big)^2\big].$$

We also need the following lemma, which gives a trivial bound of KL-regularized objective.

**Lemma B.2.** *Let $r : \mathcal{A} \to [0,1]$ be any reward function and $\pi(a) \propto \pi^{\mathsf{ref}}(a) \exp\big(\eta \cdot r(a)\big)$ be the corresponding optimal policy, then we have $J(\pi^*) - J(\pi) \le 1$.*

---

[5]The fact that $N_t(a)$ is itself a random variable seemingly prevents the application of Hoeffding's inequality, which is also a standard caveat; we refer the readers to, e.g., Orabona (2019, Section 11.2.3) for details.

*Proof of Lemma B.2.* By Lemma D.4, we know that $J(\pi^*) - J(\pi) = \eta^{-1}\mathsf{KL}(\pi\|\pi^*)$. Also, $\log(\pi/\pi^*) \leq \eta$ (Wu et al., 2025a, Lemma 1). Combining the two bounds finishes the proof. $\qquad\square$

Now we are ready to prove the "fast rate" upper bound. On the high-probability event $\mathcal{E}(\delta)$, we can decompose the regret by

$$
\begin{aligned}
\mathrm{Regret}(T) &= \sum_{t=1}^{T} \left[ J(\pi^*) - J(\pi_t) \right] \\
&\leq \sum_{t=1}^{T} \eta \mathbb{E}_{a\sim\pi_t} \left[ \left( \widehat{r}_t(a) - r(a) \right)^2 \right] \\
&\leq \eta \sum_{t=1}^{T} \mathbb{E}_{a\sim\pi_t} \left[ \frac{8\log(KT/\delta)}{N_{t-1}(a) \vee 1} \right],
\end{aligned} \tag{B.1}
$$

where the first inequality holds due to Lemma B.1 and the second inequality is by the definition of event $\mathcal{E}(\delta)$. To obtain a high-probability upper bound for $\mathrm{Regret}(T)$, we conduct the following decomposition

$$
\sum_{t=1}^{T} \mathbb{E}_{a\sim\pi_t} \left[ \frac{1}{N_{t-1}(a) \vee 1} \right] = \underbrace{\sum_{t=1}^{T} \left[ \frac{1}{N_{t-1}(a_t) \vee 1} \right]}_{I_1} + \underbrace{\sum_{t=1}^{T} \left( \mathbb{E}_{a\sim\pi_t} \left[ \frac{1}{N_{t-1}(a) \vee 1} \right] - \frac{1}{N_{t-1}(a_t) \vee 1} \right)}_{I_2} \tag{B.2}
$$

For $I_1$, we can bound it as follows:

$$
\begin{aligned}
I_1 &= \sum_{t=1}^{T} \left[ \frac{1}{N_{t-1}(a_t) \vee 1} \right] \\
&= \sum_{a\in\mathcal{A}} \left( 1 + \sum_{i=1}^{N_{T-1}(a)} \frac{1}{i} \right) \\
&\leq \sum_{a\in\mathcal{A}} \left( 1 + \sum_{i=1}^{N_{T-1}(a)} 2\log\left(1 + \frac{1}{i}\right) \right),
\end{aligned} \tag{B.3}
$$

where the last inequality holds due to $x \leq 2\log(1+x)$ when $0 < x \leq 1$. To move on, we have

$$
\begin{aligned}
\sum_{a\in\mathcal{A}} \left( 1 + \sum_{i=1}^{N_{T-1}(a)} 2\log\left(1 + \frac{1}{i}\right) \right) &= \sum_{a\in\mathcal{A}} \left( 1 + \sum_{i=1}^{N_{T-1}(a)} 2\log\left(\frac{i+1}{i}\right) \right) \\
&= \sum_{a\in\mathcal{A}} \left( 1 + 2\log\left[ \prod_{i=1}^{N_{T-1}(a)} \frac{i+1}{i} \right] \right) \\
&= \sum_{a\in\mathcal{A}} \left( 1 + 2\log\left( N_{T-1}(a) + 1 \right) \right) \\
&\leq 4K\log T,
\end{aligned} \tag{B.4}
$$

where the last inequality holds due to $N_{T-1}(a) \leq T, \forall a \in \mathcal{A}$. Thus, we know $I_1 \leq 4K\log T$. For $I_2$, let $x_t = \left( \mathbb{E}_{a\sim\pi_t} \left[ 1/(N_{t-1}(a) \vee 1) \right] - 1/(N_{t-1}(a_t) \vee 1) \right)$. Let $\mathcal{F}_t = \sigma(a_1, r_1, a_2, r_2, \ldots, a_t, r_t)$ be the $\sigma$-algebra generated by the actions and rewards up to time $t$. Then, we know $x_t$ is $\mathcal{F}_t$-measurable and $\mathbb{E}[x_t|\mathcal{F}_{t-1}] = 0$.

Let $\mathcal{E}_i(\tau) = \left\{ \sum_{t=1}^{\tau} \mathbb{E}_{a\sim\pi_t} \left[ 1/(N_{t-1}(a) \vee 1) \right] \leq 2^i \right\}$. Then, $\mathbb{1}\left( \mathcal{E}_i(t) \right)$ is $\mathcal{F}_{t-1}$-measurable. Thus, $\mathbb{E}[x_t \mathbb{1}\left( \mathcal{E}_i(t) \right)|\mathcal{F}_{t-1}] = \mathbb{1}\left( \mathcal{E}_i(t) \right)\mathbb{E}[x_t|\mathcal{F}_{t-1}] = 0$. Moreover, we have

$$
\begin{aligned}
\mathbb{E}\left[ \left( x_t \mathbb{1}[\mathcal{E}_i(t)] \right)^2 | \mathcal{F}_{t-1} \right] &= \mathbb{E}\left[ x_t^2 \mathbb{1}[\mathcal{E}_i(t)] | \mathcal{F}_{t-1} \right] \\
&= \mathbb{1}[\mathcal{E}_i(t)] \cdot \mathbb{E}\left[ \left( \mathbb{E}_{a\sim\pi_t} \left[ \frac{1}{N_{t-1}(a) \vee 1} \right] - \frac{1}{N_{t-1}(a_t) \vee 1} \right)^2 \Big| \mathcal{F}_{t-1} \right]
\end{aligned}
$$

$$= \mathbb{1}[\mathcal{E}_i(t)] \cdot \left( \mathbb{E}_{a \sim \pi_t}\left[ \left(\frac{1}{N_{t-1}(a_t) \vee 1}\right)^2 \right] - \left( \mathbb{E}_{a \sim \pi_t}\left[ \frac{1}{N_{t-1}(a) \vee 1}\right]\right)^2 \right)$$

$$\leq \mathbb{1}[\mathcal{E}_i(t)] \cdot \mathbb{E}_{a \sim \pi_t}\left[ \left(\frac{1}{N_{t-1}(a_t) \vee 1}\right)^2 \right]$$

$$\leq \mathbb{1}[\mathcal{E}_i(t)] \cdot \mathbb{E}_{a \sim \pi_t}\left[ \frac{1}{N_{t-1}(a_t) \vee 1}\right],$$

where the first inequality holds as we drop the nonpositive term. The second inequality holds due to $1/(N_{t-1}(a_t) \vee 1) \leq 1$. Therefore, we have

$$\sum_{s=1}^{t} \mathbb{E}\left[ \left( x_s \, \mathbb{1}[\mathcal{E}_i(s)]\right)^2 | \mathcal{F}_{s-1}\right] \leq \sum_{s=1}^{t} \mathbb{1}[\mathcal{E}_i(s)] \cdot \mathbb{E}_{a \sim \pi_s}\left[ \frac{1}{N_{s-1}(a_s) \vee 1}\right].$$

Let $\tau_i := \max\left\{ t \in [T] : \sum_{s=1}^{t} \mathbb{E}_{a \sim \pi_s}\left[ 1/(N_{s-1}(a) \vee 1)\right] \leq 2^i \right\}$. If $t \leq \tau_i$, $\mathbb{1}(\mathcal{E}_i(s)) = 1$ for any $s \leq t$; which means

$$t \leq \tau_i \implies \sum_{s=1}^{t} \mathbb{1}[\mathcal{E}_i(s)] \cdot \mathbb{E}_{a \sim \pi_s}\left[ \frac{1}{N_{s-1}(a_s) \vee 1}\right] = \sum_{s=1}^{t} \mathbb{E}_{a \sim \pi_s}\left[ \frac{1}{N_{s-1}(a_s) \vee 1}\right] \leq 2^i. \tag{B.5}$$

The inequality holds due to $\mathbb{1}(\mathcal{E}_i(t)) = 1$. Otherwise, if $t > \tau_i$, we have

$$t > \tau_i \implies \sum_{s=1}^{t} \mathbb{1}[\mathcal{E}_i(s)] \cdot \mathbb{E}_{a \sim \pi_s}\left[ \frac{1}{N_{s-1}(a_s) \vee 1}\right] = \sum_{s=1}^{\tau_i} \mathbb{1}[\mathcal{E}_i(s)] \mathbb{E}_{a \sim \pi_s}\left[ \frac{1}{N_{s-1}(a_s) \vee 1}\right]$$

$$+ \sum_{s=\tau_i+1}^{t} \mathbb{E}_{a \sim \pi_s} \mathbb{1}[\mathcal{E}_i(s)]\left[ \frac{1}{N_{s-1}(a_s) \vee 1}\right]$$

$$= \sum_{s=1}^{\tau_i} \mathbb{E}_{a \sim \pi_s}\left[ \frac{1}{N_{s-1}(a_s) \vee 1}\right]$$

$$\leq 2^i, \tag{B.6}$$

where we use $\mathbb{1}(\mathcal{E}_i(s)) = 1, \forall s \leq \tau_i$ and $\mathbb{1}(\mathcal{E}_i(s)) = 0, \forall s > \tau_i$; the last inequality holds due to $\mathbb{1}(\mathcal{E}_i(\tau_i)) = 1$. Therefore, we always have

$$\sum_{s=1}^{t} \mathbb{E}\left[ \left( x_s \, \mathbb{1}[\mathcal{E}_i(s)]\right)^2 | \mathcal{F}_{s-1}\right] \leq 2^i.$$

Using Freedman's inequality (Lemma D.2), we have for any $i$, with probability at least $1 - \delta/(\lceil \log_2 T \rceil)$, the following inequality holds:

$$- \sum_{t=1}^{T} \frac{1}{N_{t-1}(a_t) \vee 1} \cdot \mathbb{1}(\mathcal{E}_i(t)) + \sum_{t=1}^{T} \mathbb{E}_{a \sim \pi_t}\left[ \frac{1}{N_{t-1}(a) \vee 1} \mathbb{1}(\mathcal{E}_i(t))\right]$$

$$\leq \sqrt{2 \cdot 2^i \log(\lceil \log T \rceil/\delta)} + 2/3 \cdot \log(\lceil \log T \rceil/\delta).$$

Taking the union bound, we have with probability at least $1 - \delta$, the above inequality holds for any $1 \leq i \leq \lceil \log_2 T \rceil$. We take $i = \lceil \log_2 \sum_{t=1}^{T} \mathbb{E}_{a \sim \pi_t}[1/(N_{t-1}(a) \vee 1)]\rceil \leq \lceil \log T \rceil$. Then, $\mathbb{1}(\mathcal{E}_i(t)) = 1$ holds for any $t \leq T$. This gives us

$$I_2 = - \sum_{t=1}^{T} \frac{1}{N_{t-1}(a_t) \vee 1} + \sum_{t=1}^{T} \mathbb{E}_{a \sim \pi_t}\left[ \frac{1}{N_{t-1}(a) \vee 1}\right]$$

$$\leq \sqrt{4 \cdot \sum_{t=1}^{T} \mathbb{E}_{a \sim \pi_t}\left[ \frac{1}{N_{t-1}(a) \vee 1}\right] \cdot \log(\lceil \log T \rceil/\delta)} + 2/3 \cdot \log(\lceil \log T \rceil/\delta). \tag{B.7}$$

Substituting (B.4) and (B.7) into (B.2), we have

$$\sum_{t=1}^{T} \mathbb{E}_{a \sim \pi_t} \left[ \frac{1}{N_{t-1}(a) \vee 1} \right] \leq \sqrt{4 \cdot \sum_{t=1}^{T} \mathbb{E}_{a \sim \pi_t} \left[ \frac{1}{N_{t-1}(a) \vee 1} \right] \cdot \log(\lceil \log T \rceil / \delta)} + 4K \log T + 2/3 \cdot \log(\lceil \log T \rceil / \delta).$$

Using $x \leq a\sqrt{x} + b \Rightarrow x \leq a^2 + 2b$, we have

$$\sum_{t=1}^{T} \mathbb{E}_{a \sim \pi_t} \left[ \frac{1}{N_{t-1}(a) \vee 1} \right] \leq 6 \log(\lceil \log T \rceil / \delta) + 8K \log T. \tag{B.8}$$

Substituting (B.8) into (B.1), we know that with probability at least $1 - 2\delta$, the following inequality holds:

$$\text{Regret}(T) \leq 8\eta \log(KT/\delta) \Big[ 6 \log(\lceil \log T \rceil / \delta) + 4K \log T \Big]$$
$$\leq O\big( \eta K \cdot \log^2(KT/\delta) \big).$$

In the next step, we consider the slow rate. Still, the following proof is conditioned on $\mathcal{E}(\delta)$. The regret can be decomposed as follows:

$$\text{Regret}(T) = \sum_{t=1}^{T} \left[ \mathbb{E}_{a \sim \pi^*} \left[ r(a) - \eta^{-1} \log \frac{\pi^*(a)}{\pi^{\text{ref}}(a)} \right] - \mathbb{E}_{a \sim \pi_t} \left[ r(a) - \eta^{-1} \log \frac{\pi_t(a)}{\pi^{\text{ref}}(a)} \right] \right]$$
$$\leq \sum_{t=1}^{T} \left[ \mathbb{E}_{a \sim \pi^*} \left[ \widehat{r}_t(a) - \eta^{-1} \log \frac{\pi^*(a)}{\pi^{\text{ref}}(a)} \right] - \mathbb{E}_{a \sim \pi_t} \left[ r(a) - \eta^{-1} \log \frac{\pi_t(a)}{\pi^{\text{ref}}(a)} \right] \right]$$
$$\leq \sum_{t=1}^{T} \left[ \mathbb{E}_{a \sim \pi_t} \left[ \widehat{r}_t(a) - \eta^{-1} \log \frac{\pi^*(a)}{\pi^{\text{ref}}(a)} \right] - \mathbb{E}_{a \sim \pi_t} \left[ r(a) - \eta^{-1} \log \frac{\pi_t(a)}{\pi^{\text{ref}}(a)} \right] \right]$$
$$= \sum_{t=1}^{T} \mathbb{E}_{a \sim \pi_t} [\widehat{r}_t(a) - r(a)]$$
$$\leq \sum_{t=1}^{T} \mathbb{E}_{a \sim \pi_t} \left[ 2\sqrt{\frac{2 \log(TK/\delta)}{N_{t-1}(a) \vee 1}} \right], \tag{B.9}$$

where the first inequality holds due to $\widehat{r}_t$ is optimistic on event $\mathcal{E}(\delta)$, the second inequality holds due to $\pi_t$ is optimal under $\widehat{r}_t$ and the last inequality holds on event $\mathcal{E}(\delta)$. We have

$$\sum_{t=1}^{T} \mathbb{E}_{a \sim \pi_t} \left[ \frac{1}{\sqrt{N_{t-1}(a) \vee 1}} \right] = \underbrace{\sum_{t=1}^{T} \left[ \frac{1}{\sqrt{N_{t-1}(a_t) \vee 1}} \right]}_{J_1} + \underbrace{\sum_{t=1}^{T} \left( \mathbb{E}_{a \sim \pi_t} \left[ \frac{1}{\sqrt{N_{t-1}(a) \vee 1}} \right] - \frac{1}{\sqrt{N_{t-1}(a_t) \vee 1}} \right)}_{J_2}. \tag{B.10}$$

For $J_1$, we have

$$\sum_{t=1}^{T} \left[ \frac{1}{\sqrt{N_{t-1}(a_t) \vee 1}} \right] = \sum_{a \in \mathcal{A}} \left[ 1 + \sum_{i=1}^{N_{T-1}(a)} \frac{1}{\sqrt{i}} \right]$$
$$\leq \sum_{a \in \mathcal{A}} \left[ 2 + \int_{u=1}^{N_{T-1}(a)} \frac{1}{\sqrt{u}} du \right]$$
$$= \sum_{a \in \mathcal{A}} \left[ \frac{3}{2} + \frac{\sqrt{N_{T-1}(a)}}{2} \right]$$
$$\leq 2K + \sum_{a \in \mathcal{A}} \sqrt{N_{T-1}(a)}$$

$$\leq 2K + \sqrt{K \sum_{a \in \mathcal{A}} N_{T-1}(a)}$$

$$\leq 2K + \sqrt{KT}, \tag{B.11}$$

where the first inequality holds due to $1/\sqrt{i} \leq \int_{i-1}^{i}(1/\sqrt{u})du$. The second inequality is trivial. The third inequality holds due to the Jensen's inequality. The last inequality holds due to $\sum_{a \in \mathcal{A}} N_{T-1}(a) = T - 1$. For $J_2$, we apply Lemma D.3. Then, with probability at least $1 - \delta$, we have

$$J_2 = \sum_{t=1}^{T} \left( \mathbb{E}_{a \sim \pi_t} \left[ \frac{1}{\sqrt{N_{t-1}(a) \vee 1}} \right] - \frac{1}{\sqrt{N_{t-1}(a_t) \vee 1}} \right)$$

$$\leq 2\sqrt{2T \log(1/\delta)}. \tag{B.12}$$

Substituting (B.11) and (B.12) into (B.10), we have

$$\sum_{t=1}^{T} \mathbb{E}_{a \sim \pi_t} \left[ \frac{1}{\sqrt{N_{t-1}(a) \vee 1}} \right] \leq 2K + \sqrt{KT} + 2\sqrt{2T \log(1/\delta)}.$$

Combining this with (B.9), we have with probability at least $1 - 2\delta$,

$$\text{Regret}(T) \leq 2\sqrt{2 \log(TK/\delta)} \left[ 2K + \sqrt{KT} + 2\sqrt{2T \log(1/\delta)} \right]$$

$$\leq \widetilde{O}(K + \sqrt{KT}),$$

which finishes the proof of the regret upper bound under low-regularization. $\square$

## C  Missing Proof in Section 5

### C.1  Proof of Theorem 5.1

*Proof of Theorem 5.1.* The construction of instances follows Lattimore & Szepesvári 2020, Chapter 15. We fix $\eta$, $K \geq 9$, $T$, let $\mathcal{A} = [K]$ and select $\pi^{\text{ref}} = \text{Unif}(\mathcal{A})$. Given any reward function $r : \mathcal{A} \to [0, 1]$, we also use $r$ to denote the corresponding bandit instance $([K], r, \eta, \pi^{\text{ref}}, T)$ when there is no ambiguity. Now we take $r_1 : \mathcal{A} \to [0, 1]$ and $r_1(i) = \delta \mathbb{1}\{i = 1\}$, where $\delta > 0$ is some parameter to be figure out later. Given fixed algorithm Alg, we use $\mathbb{P}_1$ and $\mathbb{E}_1$ to denote the trajectory distribution jointly given by $r_1$ and Alg. Recall that $N_t(j)$ is the count of times the $j$-th arm has been pulled up to step $t$. Let

$$i_1 = \underset{j > 1}{\arg\min} \, \mathbb{E}_1[N_T(j)].$$

Without loss of generality, we assume $i_1 = 2$. By the pigeonhole principle, we know that $\mathbb{E}_1[N_T(2)] \leq T/(K-1)$. Now we consider the second instance given by $r_2 : \mathcal{A} \to [0, 1]$, such that $r_2(2) = 2\delta$ and $r_2(j) = r_1(j)$ for all $j \neq 2$. We now compute $\pi_1^*$ and $\pi_2^*$, which are the optimal policies under $r_1$ and $r_2$. Direct computation gives

$$\pi_1^*(1) = \frac{\exp(\eta\delta)}{\exp(\eta\delta) + K - 1}, \text{ and } \pi_1^*(i) = \frac{1}{\exp(\eta\delta) + K - 1} \text{ for all } i > 1,$$

and

$$\pi_2^*(1) = \frac{\exp(\eta\delta)}{\exp(\eta\delta) + \exp(2\eta\delta) + K - 2}, \quad \pi_2^*(2) = \frac{\exp(2\eta\delta)}{\exp(\eta\delta) + \exp(2\eta\delta) + K - 2},$$

and

$$\pi_2^*(i) = \frac{1}{\exp(\eta\delta) + \exp(2\eta\delta) + K - 2}, \ \forall \, i > 2.$$

For any policy $\pi \in \Delta(\mathcal{A})$, we consider the suboptimality gap $\text{SubOpt}_{r_1}(\pi, \pi_1^*) + \text{SubOpt}_{r_2}(\pi, \pi_2^*)$. For simplicity, we use $\text{SubOpt}_1(\pi)$ to denote $\text{SubOpt}_{r_1}(\pi, \pi_1)$ and $\text{SubOpt}_2(\pi)$ for $\text{SubOpt}_{r_2}(\pi, \pi_i)$, correspondingly. By Lemma D.4,

$$\text{SubOpt}_1(\pi) + \text{SubOpt}_2(\pi) = \eta^{-1} \left[ \text{KL}(\pi \| \pi_1^*) + \text{KL}(\pi \| \pi_2^*) \right]. \tag{C.1}$$

It is known that the unique minimizer of Equation (C.1) is $\widehat{\pi}(a) \propto \sqrt{\pi_1^*(a)\pi_2^*(a)}$ (Zhao et al., 2026, (B.9)), which gives

$$\widehat{\pi}(1) = \widehat{\pi}(2) \propto \exp(\eta\delta), \quad \text{and} \quad \widehat{\pi}(i) \propto 1, \quad \forall\, i > 2.$$

Therefore, we know that

$$\eta\big(\mathrm{SubOpt}_1(\widehat{\pi}) + \mathrm{SubOpt}_2(\widehat{\pi})\big) = \log \frac{(\exp(\eta\delta) + K - 1)(\exp(\eta\delta) + \exp(2\eta\delta) + K - 2)}{(2\exp(\eta\delta) + K - 2)^2}.$$

Now we select $\delta = \sqrt{2K/T}$. Then by the fact that $T \leq \eta^2 K/\log^2 K$, we have $\eta\delta \geq 2\log K$, and consequently $e^{\eta\delta} \geq K$, which gives that

$$\begin{aligned}
\mathrm{SubOpt}_1(\widehat{\pi}) + \mathrm{SubOpt}_2(\widehat{\pi}) &\geq \eta^{-1} \log \frac{(\exp(\eta\delta) + K - 1)(\exp(\eta\delta) + \exp(2\eta\delta) + K - 2)}{(2\exp(\eta\delta) + K - 2)^2} \\
&\geq \eta^{-1} \log \frac{\exp(2\eta\delta)(1 + \exp(\eta\delta))}{9\exp(2\eta\delta)} \\
&\geq \eta^{-1}(\eta\delta - \log 9) \\
&\geq \delta/2,
\end{aligned}$$

where the second inequality holds due to $9 \leq K \leq e^{\eta\delta}$ and the last inequality holds due to $\log 9 \leq \log K \leq \eta\delta/2$. Now, applying Lemma D.5, we obtain that

$$\inf_{\mathsf{Alg}} \sup_{r \in \mathcal{R}} \mathbb{E}_{\mathcal{D} \sim \mathbb{P}_{r,\mathsf{Alg}}}\big[\mathrm{Regret}(T)\big] \geq \frac{T\delta}{8} \cdot \exp\Big(-\mathsf{KL}\,(\mathbb{P}_1\|\mathbb{P}_2)\Big). \tag{C.2}$$

where we recall that $\mathbb{P}_{r,\mathsf{Alg}}$ is the trajectory distribution of $\mathsf{Alg}$ interacting with instance $r$, and $\mathbb{P}_\ell := \mathbb{P}_{r_\ell,\mathsf{Alg}}$. By the divergence decomposition lemma (Lattimore & Szepesvári, 2020, Lemma 15.1),

$$\mathsf{KL}\,(\mathbb{P}_1\|\mathbb{P}_2) = \sum_{k=1}^{K} \mathbb{E}_1[N_T(k)]\mathsf{KL}(r_1(k), r_2(k)) = \mathbb{E}_1[N_T(2)]\mathsf{KL}(0, 2\delta) \leq \frac{2T\delta^2}{K-1},$$

where the inequality holds due to $\mathbb{E}_1[N_T(2)] \leq T/(K-1)$ and Lemma D.1. Recall that $\delta = \sqrt{2K/T}$, we know that $\mathsf{KL}\,(\mathbb{P}_1\|\mathbb{P}_2) \leq 2K/(K-1) \leq 4.5$. Substituting them into Equation (C.2), we obtain

$$\inf_{\mathsf{Alg}} \sup_{r \in \mathcal{R}} \mathbb{E}_r\mathrm{Regret}(T) = \Omega(\sqrt{KT}),$$

where $\mathbb{E}_r$ denotes the expectation with respect to the trajectory distribution induced by $\mathsf{Alg}$ interacting with instance $r$. $\qquad\square$

## C.2  Proof of Theorem 5.3

*Proof of Theorem 5.3.* We consider the following instance class. Given $K$, $\eta$ and $\pi^{\mathsf{ref}} = \mathsf{Unif}(\mathcal{A})$ and fix some algorithm $\mathsf{Alg}$, we consider $\mathcal{A} = [2K]$ and consider a class of reward functions parameterized by some $(\mathbf{x}, \boldsymbol{\mu})$, where $\mathbf{x} \in \mathbb{R}^K$ and $\boldsymbol{\mu} \in \mathcal{V} = \{\pm 1\}^K$, such that the mean reward $r_{\mathbf{x},\boldsymbol{\mu}}(i) = 1/2 + \mathbf{x}_i + \boldsymbol{\mu}_i\delta$ for all $i \leq K$ and $r_{\mathbf{x},\boldsymbol{\mu}}(i) = 1/2 + \alpha$ for all $K < i \leq 2K$. Here $\alpha \geq 2\delta > 0$ are parameters to be assigned later *subject to* $\alpha/2\delta \in \mathbb{N}^*$. Let the reward noises follow i.i.d. standard Gaussian, which satisfy our 1-sub-Gaussian assumption on $\{\varepsilon_t\}_{t \geq 1}$. Given $\mathbf{x} \in \mathbb{R}^K$ and $\boldsymbol{\mu} \in \mathcal{V}$, we use $(\mathbf{x}, \boldsymbol{\mu})$ to denote the bandit instance $(\mathcal{A}, r_{\mathbf{x},\boldsymbol{\mu}}, \eta, \pi^{\mathsf{ref}}, T)$.

**Step 1.**  For now, we fix the first reward parameter $\mathbf{x}$ *on the premise of* $\|\mathbf{x}\|_\infty \leq \alpha - \delta$. Let $\boldsymbol{\mu}, \boldsymbol{\lambda} \in \mathcal{V}$ and consider two reward instances $(\mathbf{x}, \boldsymbol{\mu})$ and $(\mathbf{x}, \boldsymbol{\lambda})$. From now, we omit the $\mathbf{x}$ in the subscription and denote $(\mathbf{x}, \boldsymbol{\mu})$ by $\boldsymbol{\mu}$ to avoid notation clutter. Our first step is to prove that when $\eta\delta$ is small enough, for any resulted policy $\pi$, we have $\mathrm{SubOpt}_{\boldsymbol{\mu}}(\pi) + \mathrm{SubOpt}_{\boldsymbol{\lambda}}(\pi) \gtrsim \eta\delta^2 d_H(\boldsymbol{\mu}, \boldsymbol{\lambda})/K$ for all $\|\mathbf{x}\|_\infty \leq \alpha - \delta$.

We consider two instances, $\boldsymbol{\mu}_1$ and $\boldsymbol{\mu}_2$, correspondingly, such that $d_H(\boldsymbol{\mu}_1, \boldsymbol{\mu}_2) = m$, and denote the corresponding rewards by $r_1$ and $r_2$. Without loss of generality, we assume that $r_1$ and $r_2$ are given by

$$r_1(i) = 1/2 + x_i + \delta, r_2(i) = 1/2 + x_i - \delta, \forall i \in [1, l];$$

$$r_1(i) = 1/2 + x_i - \delta, r_2(i) = 1/2 + x_i + \delta, \ \forall i \in [l+1, m];$$
$$r_1(i) = r_2(i) = 1/2 + r^*(i) + x_i, r^*(i) \in \{\pm\delta\}, \ \forall i \in [m+1, K];$$
$$r_1(i) = r_2(i) = 1/2 + \alpha, \ \forall i \in [K+1, 2K],$$

where $0 \le l \le m$ and $m \le K$ are some integers. Let $\pi_1^*$ and $\pi_2^*$ be the corresponding optimal policy under rewards $r_1$ and $r_2$. For simplicity, we use $\mathrm{SubOpt}_1(\pi)$ to denote $\mathrm{SubOpt}_{r_1}(\pi, \pi_1)$ and $\mathrm{SubOpt}_2(\pi)$ for $\mathrm{SubOpt}_{r_2}(\pi, \pi_i)$, correspondingly. By Lemma D.4, we know that

$$\mathrm{SubOpt}_1(\pi) + \mathrm{SubOpt}_2(\pi) = \eta^{-1}\mathsf{KL}\left(\pi \| \pi_1^*\right) + \eta^{-1}\mathsf{KL}\left(\pi \| \pi_2^*\right).$$

Let $\widehat{\pi}$ be the minimizer of the above equation, we know that $\widehat{\pi}(i) \propto \sqrt{\pi_1^*(i)\pi_2^*(i)}$ and this gives

$$\mathrm{SubOpt}_1(\widehat{\pi}) + \mathrm{SubOpt}_2(\widehat{\pi})$$
$$= 2\eta^{-1} \log \frac{\sqrt{\sum_{i=1}^{2K} \exp(\eta r_1(i))}\sqrt{\sum_{j=1}^{2K} \exp(\eta r_2(j))}}{\sum_{k=1}^{2K} \exp\left(\eta(r_1(k) + r_2(k))/2\right)}$$
$$= \eta^{-1}\left[\underbrace{\log \frac{\sum_{i=1}^{2K} \exp(\eta r_1(i))}{\sum_{k=1}^{2K} \exp\left(\eta(r_1(k) + r_2(k))/2\right)}}_{X_1} + \underbrace{\log \frac{\sum_{i=1}^{2K} \exp(\eta r_2(i))}{\sum_{k=1}^{2K} \exp\left(\eta(r_1(k) + r_2(k))/2\right)}}_{X_2}\right].$$

The first term $X_1$ can be computed as follows

$$X_1 = \log \frac{\sum_{j=1}^{l} \exp(\eta x_j + \eta\delta) + \sum_{j=l+1}^{m} \exp(\eta x_j - \eta\delta) + \sum_{j=m+1}^{K} \exp\left(\eta x_j + \eta r^*(j)\right) + \sum_{j=K+1}^{2K} \exp(\eta\alpha)}{\underbrace{\sum_{j=1}^{m} \exp(\eta x_j) + \sum_{j=m+1}^{K} \exp\left(\eta x_j + \eta r^*(j)\right) + \sum_{j=K+1}^{2K} \exp(\eta\alpha)}_{M}}$$

$$= \log \frac{\sum_{j=1}^{l} \exp(\eta x_j + \eta\delta) + \sum_{j=l+1}^{m} \exp(\eta x_j - \eta\delta) + M}{\sum_{j=1}^{m} \exp(\eta x_j) + M}.$$

Similarly, we know that

$$X_2 = \log \frac{\sum_{j=1}^{l} \exp(\eta x_j - \eta\delta) + \sum_{j=l+1}^{m} \exp(\eta x_j + \eta\delta) + M}{\sum_{j=1}^{m} \exp(\eta x_j) + M}.$$

Now combining these two terms, we obtain that

$$X_1 + X_2 = \log \frac{\sum_{j=1}^{l} \exp(\eta x_j + \eta\delta) + \sum_{j=l+1}^{m} \exp(\eta x_j - \eta\delta) + M}{\sum_{j=1}^{m} \exp(\eta x_j) + M}$$
$$+ \log \frac{\sum_{j=1}^{l} \exp(\eta x_j - \eta\delta) + \sum_{j=l+1}^{m} \exp(\eta x_j + \eta\delta) + M}{\sum_{j=1}^{m} \exp(\eta x_j) + M}. \tag{C.3}$$

Notice that

$$\sum_{j=1}^{l} \exp(\eta x_j + \eta\delta) + \sum_{j=l+1}^{m} \exp(\eta x_j - \eta\delta) + \sum_{j=1}^{l} \exp(\eta x_j - \eta\delta) + \sum_{j=l+1}^{m} \exp(\eta x_j + \eta\delta)$$
$$= \sum_{j=1}^{m} \left(\exp(\eta x_j - \eta\delta) + \exp(\eta x_j + \eta\delta)\right),$$

where the RHS is independent to $l$. Therefore, by the concavity of $x \mapsto \log x$, Equation (C.3) is minimized when the two terms differ the most, i.e., $l = 0$ or $l = m$. We thus obtain

$$
\begin{aligned}
X_1 + X_2 &\geq \log \frac{\sum_{j=1}^{m} \exp(\eta x_j + \eta\delta) + M}{\sum_{j=1}^{m} \exp(\eta x_j) + M} + \log \frac{\sum_{j=1}^{m} \exp(\eta x_j - \eta\delta) + M}{\sum_{j=1}^{m} \exp(\eta x_j) + M} \\
&= \log \frac{\left(\sum_{j=1}^{m} \exp(\eta x_j)\right)^2 + M^2 + M \sum_{j=1}^{m} \exp(\eta x_j)\left(\exp(\eta\delta) + \exp(-\eta\delta)\right)}{\left(\sum_{j=1}^{m} \exp(\eta x_j) + M\right)^2} \\
&= \log \left(1 + \frac{2M}{\left(\sum_{j=1}^{m} \exp(\eta x_j) + M\right)^2} \sum_{j=1}^{m} \left(\exp(\eta x_j)\left(\frac{\exp(\eta\delta) + \exp(-\eta\delta)}{2} - 1\right)\right)\right).
\end{aligned}
$$

Now we come to bound the term $M$, which is straightforward since we have $-\alpha \leq x_j \pm \delta \leq \alpha$.

$$
m \leq K \exp(\eta\alpha) \leq M = \sum_{j=m+1}^{K} \exp\left(\eta r^*(j)\right) + K \exp(\eta\alpha) \leq 2K \exp(\eta\alpha).
$$

Therefore, we know that

$$
\frac{2M}{\left(\sum_{j=1}^{m} \exp(\eta x_j) + M\right)^2} \sum_{j=1}^{m} \exp(\eta x_j) \geq \frac{2mM \exp(-\eta\alpha)}{9K^2 \exp(2\eta\alpha)} \geq \frac{m}{5K \exp(2\eta\alpha)}.
$$

This enables us to bound the suboptimality gap as follows

$$
\begin{aligned}
&\mathrm{SubOpt}_1(\widehat{\pi}) + \mathrm{SubOpt}_2(\widehat{\pi}) \\
&\geq \eta^{-1} \log \left(1 + \frac{2M}{\left(\sum_{j=1}^{m} \exp(\eta \mathbf{x}_j) + M\right)^2} \sum_{j=1}^{m} \left(\exp(\eta x_j)\left(\frac{\exp(\eta\delta) + \exp(-\eta\delta)}{2} - 1\right)\right)\right) \\
&\geq \eta^{-1} \log \left(1 + \frac{m}{5K \exp(2\eta\alpha)} \cdot \left(\frac{\exp(\eta\delta) + \exp(-\eta\delta)}{2} - 1\right)\right) \\
&\geq \eta^{-1} \log \left(1 + \frac{m}{5K \exp(2\eta\alpha)} \eta^2 \delta^2\right),
\end{aligned}
\tag{C.4}
$$

where the last inequality holds due to $\forall x \in \mathbb{R}, (e^x + e^{-x})/2 - 1 \geq x^2/2$. By $\alpha \geq 2\delta$ and $\max_{x \geq 0} x^2 - 5e^{4x} \leq 0$, we know that $m\eta^2\delta^2 \leq 5K \exp(2\eta\alpha)$. Since $\forall x \in [0,1], \log(1 + x) \geq x/2$, we further have

$$
\text{Equation (C.4)} \geq \frac{m}{10K \exp(2\eta\alpha)} \eta \delta^2,
\tag{C.5}
$$

which finishes our first step.

**Step 2.** Let us first fix a time step $t \geq \eta^2 K$, and set $\alpha = 2\eta^{-1} \log 2$, which implies $\alpha\sqrt{t/K} \geq 1$, for all $t \geq \eta^2 K$, $\exists \delta_t \in [0.5\sqrt{K/t}, \sqrt{K/t}]$ such that $\alpha/2\delta_t \in \mathbb{N}^*$. Fixing such pair of $(t, \delta_t)$ and setting $\delta = \delta_t$ in Equation (C.5) yields that, for any policy $\pi$ and $\mathbf{x} \in [-\alpha + \delta_t, \alpha - \delta_t]^K$,

$$
\begin{aligned}
\mathbb{E}_{\boldsymbol{\mu} \sim \mathsf{Unif}(\mathcal{V})} \mathbb{E}_{\boldsymbol{\mu}, t}\left[\mathrm{SubOpt}_{(\mathbf{x}, \boldsymbol{\mu}), t}(\pi)\right] &\geq \frac{\eta\delta_t^2}{10^3 K} \sum_{j=1}^{K} \frac{1}{2|\mathcal{V}|} \sum_{\boldsymbol{\mu} \sim_j \boldsymbol{\lambda}} \exp\left(-\mathsf{KL}(\mathbb{P}_{\boldsymbol{\mu}, t} \| \mathbb{P}_{\boldsymbol{\lambda}, t})\right) \\
&= \frac{\eta\delta_t^2}{2^{11}|\mathcal{V}|K} \sum_{d_H(\boldsymbol{\mu}, \boldsymbol{\lambda})=1} \exp\left(-\mathsf{KL}(\mathbb{P}_{\boldsymbol{\mu}, t} \| \mathbb{P}_{\boldsymbol{\lambda}, t})\right) \\
&\geq \frac{\eta\delta_t^2}{2^{10}} \exp\left(-\frac{1}{2|\mathcal{V}|K} \sum_{d_H(\boldsymbol{\mu}, \boldsymbol{\lambda})=1} \mathsf{KL}(\mathbb{P}_{\boldsymbol{\mu}, t} \| \mathbb{P}_{\boldsymbol{\lambda}, t})\right),
\end{aligned}
$$

where the first inequality is by plugging Equation (C.5) into Lemma D.6, and the last inequality holds due to Jensen's inequality.[6] Then for any fixed $\boldsymbol{\mu}$, the standard divergence decomposition lemma (Lattimore & Szepesvári, 2020, Lemma 15.1) gives

$$\sum_{\boldsymbol{\lambda}:d_H(\boldsymbol{\mu},\boldsymbol{\lambda})=1} \mathsf{KL}(\mathbb{P}_{\boldsymbol{\mu},t}\|\mathbb{P}_{\boldsymbol{\lambda},t}) = \sum_{k=1}^{K} \mathbb{E}_{\boldsymbol{\mu},t}[N_t(k)]\mathsf{KL}(+\delta_t,-\delta_t) = 2t\delta_t^2,$$

where we recall that $\mathsf{KL}(+\delta_t\|-\delta_t) = \mathsf{KL}(1/2 + \mathbf{x}_j + \delta_t\|1/2 + \mathbf{x}_j - \delta_t) = 2\delta_t^2$ denotes the KL divergence from $\mathcal{N}(1/2 + \mathbf{x}_j + \delta_t, 1)$ to $\mathcal{N}(0.5 + x_j - \delta_t, 1)$ and happens to be symmetric Lemma D.1. Therefore, we know that

$$\mathbb{E}_{\boldsymbol{\mu}\sim\mathsf{Unif}(\mathcal{V})}\mathbb{E}_{\boldsymbol{\mu},t}\big[\mathrm{SubOpt}_{(\mathbf{x},\boldsymbol{\mu}),t}(\pi)\big] \geq \frac{\eta\delta_t^2}{2^{10}} \exp\left(-\frac{1}{2|\mathcal{V}|K}\sum_{d_H(\boldsymbol{\mu},\boldsymbol{\lambda})=1}\mathsf{KL}(\mathbb{P}_{\boldsymbol{\mu},t}\|\mathbb{P}_{\boldsymbol{\lambda},t})\right)$$
$$\geq \frac{\eta\delta_t^2}{2^{10}}\exp\left(-\frac{t\delta_t^2}{K}\right).$$
$$\geq \frac{\eta K}{2^{10}t}\exp(-1),$$

where the last inequality holds due to $\delta_t \in [\sqrt{K/t}/2, \sqrt{K/t}]$. Recall that $N_t := \alpha/2\delta_t$ is a positive integer by design, we define $\mathcal{H}_t := \cup_{j=1}^{N_t}[-\alpha + (4j-3)\delta_t, -\alpha + (4j-1)\delta_t]$, then we notice that if we take $\mathbf{x} \sim \mathsf{Unif}(\mathcal{H}_t^K)$ and $\boldsymbol{\mu} \sim \mathsf{Unif}(\mathcal{V})$ independently, then $\mathbf{x} + \boldsymbol{\mu}\delta_t \sim \mathsf{Unif}([-\alpha, \alpha]^K)$. Therefore, the tower property gives

$$\mathbb{E}_{(r_{[1:K]}-1/2)\sim\mathsf{Unif}([-\alpha,\alpha]^K)}\mathbb{E}_{\boldsymbol{\mu},t}\big[\mathrm{SubOpt}_{r,t}(\pi)\big] = \mathbb{E}_{\mathbf{x}\sim\mathsf{Unif}(\mathcal{H}_t)}\mathbb{E}_{\boldsymbol{\mu}\sim\mathsf{Unif}(\mathcal{V})}\mathbb{E}_{\boldsymbol{\mu},t}\big[\mathrm{SubOpt}_{(\mathbf{x},\boldsymbol{\mu}),t}(\pi)\big] \geq \frac{\eta K}{2^{12}t}, \quad\text{(C.6)}$$

where $r_{[1:K]}$ denotes the first $K$ coordinates of the mean reward function $r_{\mathbf{x},\boldsymbol{\mu}}$ (See Figure 4 for an intuitive illustration of the equality in Equation (C.6)). Invoking the tower property again yields that for any policy $\pi$,

$$\sup_r \mathbb{E}_{(\pi,r)}\mathrm{Regret}_r(T) \geq \mathbb{E}_{(r_{[1:K]}-1/2)\sim\mathsf{Unif}([-\alpha,\alpha]^K)}[\mathbb{E}_{(\pi,r)}\mathrm{Regret}_r(T)]$$
$$\geq \sum_{t=\lceil\eta^2 K\rceil}^{T} \mathbb{E}_{(\mathbf{x},\boldsymbol{\mu})\sim\mathsf{Unif}(\mathcal{H}_t\times\mathcal{V})}\mathbb{E}_{\boldsymbol{\mu},t}\big[\mathrm{SubOpt}_{(\mathbf{x},\boldsymbol{\mu}),t}(\pi)\big] \geq 2^{-12}\eta K \sum_{t=\lceil\eta^2 K\rceil}^{T} t^{-1}, \quad\text{(C.7)}$$

where Equation (C.7) follows from Equation (C.6). Finally, $\sum_{t=\lceil\eta^2 K\rceil}^{T} t^{-1} = \Omega\big(\log(T/\eta^2 K)\big)$ concludes the proof. $\square$

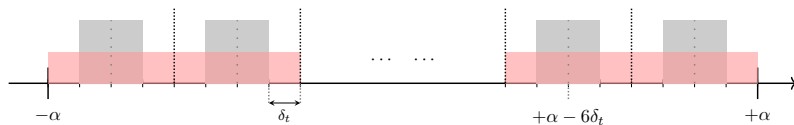

*Figure 4.* The shared uniform Bayes prior for every $t \geq \eta^2 K$. The plot above takes 1 out of $K$ axes of $\mathsf{Unif}\big([-\alpha, +\alpha]^K\big)$ for illustration. The gray boxes denote the density of $\mathbf{x}$ and hence the red boxes represent the density of $\mathbf{x} + \boldsymbol{\mu}\delta_t$.

*Remark* C.1. In general, a regret lower bound cannot be converted to a sample complexity lower bound by directly applying the online-to-batch conversion. Nevertheless, The proof of Theorem 5.3 indeed implies a $\Omega(\eta K/\epsilon)$ sample complexity lower bound. In particular, if we take $t = T$ in (C.6) and then take maximum over all possible reward functions in the support of this Bayesian prior, then we obtain that, for any algorithm

$$\sup_r \mathbb{E}_{(\pi,r)}\big[\mathrm{SubOpt}_{r,t}(\pi)\big] \gtrsim \frac{\eta K}{T},$$

This leads to the $\Omega(\eta K/\epsilon)$ sample complexity lower bound.

---

[6]The notation $\mathbb{E}_{\boldsymbol{\mu},t}[\cdot]$ is with respect to the trajectory distribution of the interaction between $\pi$ and the instance $\boldsymbol{\mu}$ up to time step $t$.

# D    Auxiliary Lemmas

We first recall a standard fact about the KL divergence between two Gaussian distributions with unit variance.

**Lemma D.1.** $\forall m, \delta \in \mathbb{R}, \mathsf{KL}(m, m + 2\delta) \coloneqq \mathsf{KL}\left(\mathcal{N}(m, 1) \| \mathcal{N}(m + 2\delta, 1)\right) = 2\delta^2$.

**Lemma D.2** (Freedman's inequality, Freedman 1975). *Let $M, v > 0$ be fixed constants. Let $\{x_i\}_{i=1}^n$ be a stochastic process, $\{\mathcal{F}_i\}_i$ be a filtration so that for $i \in [n], x_i$ is $\mathcal{F}_i$-measurable, while almost surely*

$$\mathbb{E}[x_i | \mathcal{F}_{i-1}] = 0, |x_i| \leq M, \sum_{i=1}^{n} \mathbb{E}\left[x_i^2 | \mathcal{F}_{i-1}\right] \leq v.$$

*Then for any $\delta > 0$, with probability at least $1 - \delta$, we have*

$$\sum_{i=1}^{n} x_i \leq \sqrt{2v \log(1/\delta)} + 2/3M \log(1/\delta).$$

**Lemma D.3** (Azuma-Hoeffding inequality, Azuma 1967; Cesa-Bianchi & Lugosi 2006). *Let $\{x_i\}_{i=1}^n$ be a martingale difference sequence with respect to a filtration $\{\mathcal{G}_i\}$ satisfying $|x_i| \leq M$ for some constant $M$, $x_i$ is $\mathcal{G}_{i+1}$-measurable, $\mathbb{E}[x_i | \mathcal{G}_i] = 0$. Then for any $0 < \delta < 1$, with probability at least $1 - \delta$, we have*

$$\sum_{i=1}^{n} x_i \leq M \sqrt{2n \log(1/\delta)}.$$

**Lemma D.4** (Zhao et al. 2026, (D.10)). *Consider any $\eta > 0$, finite action set $\mathcal{A}$, and reward function $r : \mathcal{A} \to \mathbb{R}$. Let $\pi^{\mathsf{ref}} \in \Delta(\mathcal{A})$ be any reference policy and $\pi^* \in \Delta(\mathcal{A})$ be the optimal policy under $r$, i.e., $\pi^*(a) \propto \pi^{\mathsf{ref}}(a) \exp(\eta r(a))$ for all $a \in \mathcal{A}$. Let $\pi$ be any policy, then the suboptimal gap between $\pi$ and $\pi^*$ under the KL-regularized objective is given by $\mathrm{SubOpt}(\pi, \pi^*) = \eta^{-1} \mathsf{KL}(\pi \| \pi^*)$.*

The following two lemmas are standard results for proving information-theoretic minimax lower bounds.

**Lemma D.5** (Le Cam's two-point method, Le Cam 1973; Yu 1997). *Let $\mathcal{R}$ be the set of instances, $\Pi$ be the set of estimators, and $L : \Pi \times \mathcal{R} \to \mathbb{R}_+$ be a loss function. For $\widetilde{r}, \bar{r} \in \mathcal{R}$, suppose $\exists c > 0$ such that*

$$\inf_{\pi \in \Pi} L(\pi, \widetilde{r}) + L(\pi, \bar{r}) \geq c,$$

*then*

$$\inf_{\pi \in \Pi} \sup_{r \in \mathcal{R}} \mathbb{E}_{\mathcal{D} \sim P_r} L\left(\pi(\mathcal{D}), r\right) \geq \frac{c}{4} \cdot \exp\left(-\mathsf{KL}\left(P_{\widetilde{r}} \| P_{\bar{r}}\right)\right),$$

*where the trajectory distribution of $\pi$ interacting with instance $r$ is denoted by $P_r$.*

We adopt the following variant of Assouad's lemma.[7]

**Lemma D.6** (Assouad's Lemma, Yu 1997). *Let $\mathcal{R}$ be the set of instances, $\Pi$ be the set of estimators, $\mathcal{V} \coloneqq \{\pm 1\}^S$ for some $S > 0$, such that $r_{\boldsymbol{\nu}} \in \mathcal{R}$ for all $\boldsymbol{\nu} \in \mathcal{V}$. Let $L : \Pi \times \mathcal{R} \to \mathbb{R}_+$ be any loss function satisfying the following separation condition*

$$L(\pi, r_{\boldsymbol{\mu}}) + L(\pi, r_{\boldsymbol{\lambda}}) \geq c \cdot d_H(\boldsymbol{\mu}, \boldsymbol{\lambda}), \quad \forall \boldsymbol{\mu}, \boldsymbol{\lambda} \in \mathcal{V} \text{ and } \pi \in \Pi$$

*for some $c \geq 0$, then for any estimator $\pi$,*

$$\mathbb{E}_{\boldsymbol{\nu} \sim \mathsf{Unif}(\mathcal{V})} \mathbb{E}_{\mathcal{D} \sim \mathbb{P}_{\boldsymbol{\nu}}} L(\pi(\mathcal{D}), r_{\boldsymbol{\nu}}) \geq \frac{c}{8|\mathcal{V}|} \sum_{j=1}^{S} \sum_{\boldsymbol{\mu} \sim_j \boldsymbol{\lambda}} \exp\left(-\mathsf{KL}(\mathbb{P}_{\boldsymbol{\mu}} \| \mathbb{P}_{\boldsymbol{\lambda}})\right),$$

*where $\boldsymbol{\mu} \sim_j \boldsymbol{\lambda}$ denotes that $d_H(\boldsymbol{\mu}, \boldsymbol{\lambda}) = 1$ and $\boldsymbol{\mu}_j \neq \boldsymbol{\lambda}_j$.*

---

[7]Similar variants have been shown in, e.g., https://theinformaticists.wordpress.com/2019/09/16/lecture-8-multiple-hypothesis-testing-tree-fano-and-assoaud

*Proof of Lemma D.6.* For any pair of policy $\pi$ and $\boldsymbol{\nu} \in \mathcal{V}$, we pick their corresponding $\widehat{\boldsymbol{\nu}} \in \operatorname{argmin}_{\boldsymbol{\nu} \in \mathcal{V}} L(\pi, r_{\boldsymbol{\nu}})$ arbitrarily to obtain

$$L(\pi, r_{\boldsymbol{\nu}}) \geq \frac{L(\pi, r_{\boldsymbol{\nu}}) + L(\pi, r_{\widehat{\boldsymbol{\nu}}})}{2} \geq \frac{c}{2} \sum_{j=1}^{S} \left( \mathbb{1}[\boldsymbol{\nu}_j = 1, \widehat{\boldsymbol{\nu}}_j = -1] + \mathbb{1}[\boldsymbol{\nu}_j = -1, \widehat{\boldsymbol{\nu}}_j = 1] \right), \forall \boldsymbol{\nu} \in \mathcal{V};$$

which in turn implies

$$\mathbb{E}_{\boldsymbol{\nu} \sim \mathsf{Unif}(\mathcal{V})} L(\pi, r_{\boldsymbol{\nu}}) \geq \frac{c}{2} \sum_{j=1}^{S} \frac{1}{|\mathcal{V}|} \left( \sum_{\boldsymbol{\nu}:\boldsymbol{\nu}_j=1} \mathbb{1}[\widehat{\boldsymbol{\nu}}_j = -1] + \sum_{\boldsymbol{\nu}:\boldsymbol{\nu}_j=-1} \mathbb{1}[\widehat{\boldsymbol{\nu}}_j = 1] \right).$$

Then for any estimator $\pi$,

$$
\begin{aligned}
\mathbb{E}_{\boldsymbol{\nu} \sim \mathsf{Unif}(\mathcal{V})} \mathbb{E}_{\mathcal{D} \sim \mathbb{P}_{\boldsymbol{\nu}}} L(\pi(\mathcal{D}), r_{\boldsymbol{\nu}}) &\geq \frac{c}{2} \sum_{j=1}^{S} \frac{1}{|\mathcal{V}|} \left( \sum_{\boldsymbol{\nu}:\boldsymbol{\nu}_j=1} \mathbb{P}_{\boldsymbol{\nu}}[\widehat{\boldsymbol{\nu}}_j = -1] + \sum_{\boldsymbol{\nu}:\boldsymbol{\nu}_j=-1} \mathbb{P}_{\boldsymbol{\nu}}[\widehat{\boldsymbol{\nu}}_j = 1] \right) \\
&= \frac{c}{2} \sum_{j=1}^{S} \frac{1}{2|\mathcal{V}|} \sum_{\boldsymbol{\mu} \sim_j \boldsymbol{\lambda}} \left( \mathbb{P}_{\boldsymbol{\mu}}(\widehat{\boldsymbol{\mu}}_j = -1) + \mathbb{P}_{\boldsymbol{\lambda}}(\widehat{\boldsymbol{\lambda}}_j = +1) \right) \\
&\geq \frac{c}{4|\mathcal{V}|} \sum_{j=1}^{S} \sum_{\boldsymbol{\mu} \sim_j \boldsymbol{\lambda}} 1 - \mathsf{TV}\left(\mathbb{P}_{\boldsymbol{\mu}} \| \mathbb{P}_{\boldsymbol{\lambda}}\right) \geq \frac{c}{8|\mathcal{V}|} \sum_{j=1}^{S} \sum_{\boldsymbol{\mu} \sim_j \boldsymbol{\lambda}} \exp\left(-\mathsf{KL}(\mathbb{P}_{\boldsymbol{\mu}} \| \mathbb{P}_{\boldsymbol{\lambda}})\right),
\end{aligned}
$$

where the penultimate inequality follows from the variational representation of TV, and the last inequality is by the Bretagnolle-Huber inequality (See e.g., Lattimore & Szepesvári (2020, Theorem 14.2)). □

