# OpenReview forum: "Near-Optimal Regret for KL-Regularized Multi-Armed Bandits"
_ICML.cc/2026/Conference — ICML 2026 regular_

### Official Review · Reviewer_A7aZ · 2026-03-07

**Soundness:** 3
**Presentation:** 3
**Significance:** 3
**Originality:** 3
**Overall Recommendation:** 5
**Confidence:** 4

**Summary:**

This paper studies stochastic $K$-armed bandits under a KL-regularized objective $J(\pi) = E_{a \sim \pi}[r(a)] - \eta^{-1}KL(\pi || \pi_{ref})$. The authors analyze a KL-UCB–style algorithm that builds optimistic reward estimates and then plays the KL-regularized optimal policy for those optimistic rewards. The main contribution is a two-regime regret characterization: in the high-regularization regime, they prove a high-probability $\tilde O(\eta K \log^2 T)$ upper bound together with a nearly matching $\Omega(\eta K \log T)$-type lower bound, while in the low-regularization regime they recover $\tilde O(\sqrt{KT} \log T)$-type behavior together with a matching lower-bound story up to logarithmic factors. Overall, the paper gives a strong and nearly complete picture of how regularization changes the statistical difficulty of bandit learning.

**Compliance With Llm Reviewing Policy:**

Affirmed.

**Final Justification:**

Most of my concerns have been adequately addressed, particularly those regarding the presentation of the theorem, the motivation for the formulation, and several experimental and technical clarifications.

**Key Questions For Authors:**

1. Are the lower bounds in Theorems 5.1 and 5.3 intended as expected-regret lower bounds? If so, please say this explicitly in the theorem statements, especially since the upper bounds are high-probability.

2. Please provide a more detail motivation to study the classic stochastic MAB under the KL-regularized objective.

3. The upper bound has $\log^2 T$ while the lower bound is $\Omega(\eta K \log(\cdot))$; the paper notes this gap, but readers may want more intuition on whether $\log^2T$ is an artifact of high-probability analysis vs inherent.

4. Do you assume $\pi_{\text{ref}}(a)>0$ for all arms? If not, how is $J(\pi)$ defined when $\pi(a)>0$ but $\pi_{\text{ref}}(a) = 0$? Do any bounds depend on $\min_{a}\pi_{\text{ref}}(a)$?

5.  Intuition for the fast rate: Can you include a short intuitive example (e.g., $K=2$) showing why strong regularization makes regret scale with a squared reward-estimation error (and hence allows logarithmic regret), and how this relates to curvature/strong convexity from the KL term?

6. Anytime/horizon-free version: The bonus uses $\log(TK/\delta)$. Can KL-UCB be made anytime (e.g., via a time-dependent bonus $\log(tK/\delta)$ or a doubling trick) without changing the rates?

7. Is $\eta$ assumed known to the learner? If $\eta$ is misspecified or unknown, can you adaptively tune it (or provide robustness guarantees)?

8. Minor clarity: There are a couple places where event notation appears inconsistent (e.g., $E(\delta)$ vs $E_1(\delta)$; please check and unify.

**Limitations:**

I do not identify any concerns regarding potential societal impact, nor do I see specific societal risks arising from this theoretical work.

**Strengths And Weaknesses:**

Strengthes:

1. Clear regime picture + near-matching lower bounds: The paper separates “weak” vs “strong” regularization and provides upper and lower bounds in both regimes, giving a fairly complete characterization of regret scaling in ($K, \eta, T$).

2. Improved dependence on $K$ in the fast-rate regime: The high-regularization bound $\tilde O(\eta K \log^2T)$ is linear in $K$, and the paper explains how this improves over bounds obtained by naively specializing more general function-approximation results.

3. Technically interesting analysis for the “fast-rate” regime. The decomposition that yields regret controlled by squared estimation error (scaled by $\eta$) makes the fast rate plausible, and the peeling argument seems like the core technical contribution.

Soundness:

The paper appears technically strong overall. The upper-bound analysis is nontrivial and the proof sketch makes the key idea fairly clear: in the strongly regularized regime, regret can be related to a squared reward-estimation error term, and the martingale/control argument based on Freedman plus peeling is what makes the logarithmic dependence plausible. The lower-bound side is also substantive, especially because the paper explains why the standard hard-instance construction is insufficient in the high-regularization regime and introduces a sharper argument to recover the $\eta K \log T$-type dependence.

My main soundness-related concerns are about statement precision rather than the core ideas.

1. The weak-regularization lower-bound story should be stated carefully: Theorem 5.1 gives $\Omega(\sqrt{KT})$ only under the stronger condition $\eta \geq \sqrt{T \log^2 K/K}$, while corollary 5.2 gives $\Omega(\sqrt{KT}/\log K)$ over the full regime $ \eta \geq \sqrt{T/K}$.

2. While a purely theoretical paper is acceptable, some experiments could help illustrate (i) the regime transition around $\eta \approx \sqrt{T/K}$, and (ii) how the learned policy $\pi_t$ behaves as $\eta$ varies (e.g., how close it stays to $\pi_{\text{ref}}$).

3. The lower-bound proofs seem to conclude expected-regret lower bounds via Le Cam–type arguments, so the theorem statements should clarify whether the bound is on expected regret.

Presentation

The paper is generally well organized and the high-level regime picture is easy to follow. The comparison to prior general-function-approximation results is also useful because it clarifies where the improved $K$-dependence is coming from.

The main presentation weaknesses are mostly about clarity and framing, detailed below.

1. The algorithmic novelty is more limited than the analytical novelty, and I think the paper would benefit from saying this more directly: the algorithm is a natural optimistic plug-in method, while the main contribution is the sharp analysis and tight regime characterization.

2. Motivation in the pure MAB setting could be stronger: The intro cites RL/RLHF-style motivations, but for classical $K$-armed bandits it would help to give one or two concrete use cases and interpret what optimizing $J(\pi)$ means operationally.

3.I would also encourage the authors to clean up a few statement-level inconsistencies, such as the switch between $E(\delta)$ and $E_1(\delta)$, and to explicitly state any support assumptions on $\pi_{ref}$, since the optimal policy is defined through this term.


Significance

I view the paper as significant within its scope. KL-regularized objectives are widely used in reinforcement learning and preference-based optimization, and the question of how regularization changes regret is an important one. The paper gives a clean regime transition and improves the dependence on $K$ in the fast-rate regime relative to what one gets by naively specializing prior general-function-approximation results to MABs. That makes the work relevant not only as a bandit result, but also as a sharp case study for KL-regularized online learning more broadly.

Originality

The originality is mainly analytical rather than algorithmic. The algorithm itself is a natural KL-UCB variant, so I would not sell the paper as introducing a fundamentally new algorithmic template. However, the paper does provide new insight into the structure of KL-regularized bandits, especially the transition between $\sqrt{T}$-type and polylogarithmic regret, the improved linear dependence on $K$ in the fast-rate regime, and the stronger lower-bound construction needed in the high-regularization case. In my view, that is a meaningful form of originality for a theory paper.

---

> ### Author Rebuttal · Authors · 2026-03-29
>
> Thank you for your constructive feedback! We answer your questions as follows.
>
> **Q1**: Theorem 5.1 gives $\Omega(\sqrt{KT})$ only under the stronger condition $\eta\geq\sqrt{T\log^2K/K}$, while Corollary 5.2 gives $\Omega(\sqrt{KT}/\log K)$ over the full regime $\eta\geq\sqrt{T/K}$.
>
> **A1**: This is by design. As we mentioned right above Corollary 5.2, it is derived from a change of variable $T \leftarrow T/\log^2K$. Its purpose is to match the phase transition threshold $\eta=\Theta(\sqrt{T/K})$ in the upper bound.
>
>
> **Q2**: Some experiments help illustrate (i) the regime transition, and (ii) how the learned policy $\pi_t$ behaves as $\eta$ varies.
>
> **A2**: Thank you for the suggestion! Experimental results about regime transitions and behavior of $\pi_t$ are compiled at https://anonymous.4open.science/api/repo/KLMAB-449C/file/Rebuttal.pdf
>
> **Q3**: The theorem statements should clarify whether the bound is for expected regret.
>
> **A3**: Thank you for your suggestion! We have made it clear in Theorem 4.2 that our upper bounds hold with high probability. We will emphasize in the revision that all our lower bounds are for the expected regret in the minimax sense.
>
>
> **Q4**: The paper would benefit from saying this more directly: the algorithm is a natural optimistic plug-in method, while the main contribution is the sharp analysis and tight regime characterization.
>
> **A4**: We refer to **W2 of Reviewer Lbos**.
>
> **Q5**: It would help to give one or two concrete use cases of $K$-armed bandits and interpret what optimizing $J(\pi)$ means operationally.
>
> **A5**: In recommendation systems, where $\pi^\text{ref}$ may encode fairness or exposure requirements (e.g., uniform exposure across providers), KL-regularization might be employed to ensure the learned policy adheres to such requirements. Here, maximizing $J(\pi)$ ensures the learned policy improves reward (e.g., user clicks) while still complying with the fairness or exposure requirements. We will add this motivating example in the revision.
>
> **Q6**:  Inconsistencies, e.g., $\mathcal E(\delta)$ vs $\mathcal E_1(\delta)$.
>
> **A6**: Thank you for pointing out these issues! We will fix them in our revision.
>
> **Q7**: Is $\log T$ gap is an artifact or inherent?
>
> **A7**: We refer to **W1 of Reviewer Lbos**.
>
>
> **Q8.1**: Do you assume $\min_{a}\pi^{\text{ref}}(a) > 0$?
>
> **A8.1**: We do not need any specific assumption on $\pi^\text{ref}$.
>
> **Q8.2**: How is $J(\pi)$ defined if $\exists a,\pi(a)>0,\pi^\text{ref}(a)=0$?
>
> **A8.2**: In this case $\mathsf{KL}(\pi\\|\pi^\text{ref})=+\infty$ and $J(\pi)=-\infty$. Actually, in the upper bound analysis, we always have $\hat{\pi}(\cdot) \propto \pi^\text{ref}(\cdot) \exp\big(\eta\hat r(\cdot)\big)$, thus $\hat\pi\ll\pi^\text{ref}$. This ensures that $\mathsf{KL}(\hat{\pi}\\|\pi^\text{ref})<\infty$. We will add this discussion in the revision.
>
> **Q8.3**: Do any bounds depend on $\min_{a}\pi^\text{ref}(a)$?
>
> **A8.3**: No. Neither our upper nor lower bounds depend on $\min_{a}\pi^\text{ref}(a)$.
>
> **Q9**: What is the connection between the fast rate and the curvature induced by the KL regularizer?
>
> **A9**: The fast rate for small $\eta$ is due to the strong convexity of $\pi\mapsto\mathsf{KL}(\pi\\| \pi^\text{ref})$ w.r.t the total variation distance (Lemma A.1). Additionally, it is worth noting that optimism of the reward estimator $\hat{r}$ is by far crucial for bounding the suboptimality from above by the "on-policy" quadratic reward estimation error. To aid intuition, consider $\pi^\text{ref}=[0.5,0.5]$, in which case $\pi^\*$ encourages uniform exploration on top of the optimistic bonus; if $\eta$ is very small (i.e., the regularization is sufficiently strong), the regularized suboptimality is essentially dominated by $\eta^{-1}\mathsf{KL}(\hat\pi \\|\pi^\*)$ (which is guaranteed by Lemma C.4). Therefore, Pinsker's inequality tells us that there should be a **quadratic** boost for $\mathsf{KL}(\hat{\pi}\\|\pi^\*)$ compared with the $L^1$ distance between $\hat\pi$ and $\pi^*$, which is exactly the source of the boosting from a $1/\sqrt{t}$ rate to a $1/t$ fast rate.
>
> **Q10**: Can KL-UCB be made anytime?
>
> **A10**: We anticipate it to be promising to incorporate the tabular-form bonus of MOSS-anytime (Degenne and Perchet, 2016) to our KL-UCB to derive anytime upper bounds of the expected regret, the details of which are left to future work.
>
> **Q11**: How to tackle unknown $\eta$?
>
> **A11**: $\eta$ is not a parameter of the model, but a hyperparameter for the objective & algorithm, and thus it is not meaningful to consider misspecified $\eta$. Also, it might be impossible to tackle unknown $\eta$: if $\eta$ is not given, the **interaction protocol** between the learner and the environment through reward query does NOT provide any information about the regularization intensity of the KL-regularized performance metric. So, it is intuitively impossible to adaptively tune it.

---

> > ### Author Rebuttal · Reviewer_A7aZ · 2026-04-04
> >
> > I thank the authors for the response. Most of my concerns were resolved. I will raise my score from 4 to 5.

---

> > > ### Author Response · Authors · 2026-04-05
> > >
> > > Thank you for your support. We are glad that our response helped address your concerns.

---

### Official Review · Reviewer_Lbos · 2026-03-08

**Soundness:** 4
**Presentation:** 3
**Significance:** 3
**Originality:** 3
**Overall Recommendation:** 4
**Confidence:** 3

**Summary:**

This paper studies stochastic KL-regularized stochastic multi-armed bandits, and analyzes the regret of online learning in this setting. This paper proposes a variant of KL-UCB and provides regret upper bounds in both the high-regularization regime and the low-regularization regime, and constructs two sets of hard instances that nearly match regret lower bounds in both regimes.

**Compliance With Llm Reviewing Policy:**

Affirmed.

**Final Justification:**

Most of my concerns are addressed and I remain my initial rating.

**Key Questions For Authors:**

1. In the bound analysis, is the extra $\log(T)$ factor inherent, or could it potentially be removed with a sharper martingale analysis?
2. The analysis critically relies on the curvature induced by reverse KL. Do the authors expect similar fast-rate regret behavior under other regularizers, such as entropy or Tsallis regularization?
3. The phase transition is around $\sqrt{T/K}$. Is this threshold information-theoretically sharp? Can you give some simple experimental results?
4. Is it possible to obtain instance-dependent regret bounds under the KL-regularized objective? Can you give some discussion?

**Limitations:**

yes.

**Strengths And Weaknesses:**

### Strengths:
1. This paper separates the regret analysis of KL-regularized MABs into two regimes. The analysis is clean and organized. In the high-regularization regime, the regret improves to $\tilde{O}(\eta K\log^2T)$. The authors further establish a matching lower bound $\Omega(\eta K\log(T))$.
2. The proof leverages the curvature of the KL-regularized objective to reduce regret to inverse visitation counts, which are controlled via double counting and a peeling technique. The hardness proof introduces a multi-coordinate instance family to overcome the failure of two-arm constructions under strong regularization. These insights are novel.
3. KL-regularized objectives are important in RL and RLHF. This paper fills the gap in understanding the statistical efficiency of online learning with these objectives in the setting of MABs.
### Weaknesses:
1. The upper and lower bounds still differ by a logarithmic factor in the high-regularization regime.
2. The algorithmic contribution is relatively modest. The algorithm is essentially a KL-UCB variant, with most novelty lying in the analysis rather than algorithm design.
3. The analysis is restricted to stochastic multi-armed bandits. The related applications such as RLHF involve more complex settings.
4.  The paper is purely theoretical and does not include empirical experiments. While this is a theory paper, simple experiments illustrating the phase transition across different regularization regimes could help improve the intuition and accessibility of the results.

Typo: 1. In line 420, algorithm 'Alg' seems mistyped.

---

> ### Author Rebuttal · Authors · 2026-03-29
>
> Thank you for your supportive feedback! We answer your questions as follows.
>
> **W1**: On the $\log T$-gap between the upper and lower bounds in the high-regularization regime.
>
> **A1**: We conjecture that the current $\log T$ gap primarily stems from the looseness of the upper bound. In particular, Theorem 4.2 provides a high-probability upper bound, which involves taking a union bound over all time steps $t\in [T]$, thus introducing an extra $\log(T)$ dependency. We conjecture that this $\log T$ gap could be removed through an analysis of the expected regret. This potentially involves adapting some minimax optimal bandit algorithms like MOSS in [1] to KL-regularized setting. We will incorporate this discussion in the revised version and leave a complete solution to this issue as future work.
>
> **W2**: The novelty mainly lies in the analysis rather than algorithm design.
>
> **A2**: Thank you for your acknowledgement of the analytical novelty. As we mentioned in the conclusion, our algorithm is a tabular variant of the algorithm in [2]. We will also emphasize this aspect in the introduction in the revision.
>
> **W3**: The analysis is restricted to stochastic multi-armed bandits. The related applications such as RLHF involve more complex settings.
>
> **A3**: The relevant predecessors out work (listed in Table 1) indeed consider more general function approximation settings in the contextual bandit framework. However, our analysis demonstrates that none of their analyses is sharp for all regimes of regularization, even specialized to MABs; in particular, the best known upper bound is loose by a factor of $K$ in the high regularization regime. Our work should be considered as the first step towards a complete understanding of more general setups of KL-regularized RLHF, etc. Finally, this is mainly a theoretical work. While we appreciate applications such as RLHF, it is beyond the focus of this paper, and we leave a thorough investigation of more complex settinga as our future works.
>
> **W4**: Simple experiments illustrating the phase transition across different regularization regimes could help improve the intuition and accessibility of the results.
>
> **A4**: Thank you for your suggestion! We conducted experiments under different $\eta$ and compiled the results at https://anonymous.4open.science/api/repo/KLMAB-449C/file/Rebuttal.pdf
>
> **W5**: Typos
>
> **A5**: Thank you for pointing out! We will revise it in our next version.
>
> **Q1**: Do the authors expect similar fast-rate regret behavior under other regularizers?
>
> **A1**: Yes. In fact, in the offline setting, [3] already showed that a similar fast rate is achievable against $f$-divergence-regularized objectives for strongly convex and twice continuously differentiable $f$. We believe that adapting these results to online setting should be possible and leave a complete investigation as our future work.
>
> **Q2**: The phase transition is around $\eta = \sqrt{T/K}$. Is this threshold information-theoretically sharp?
>
> **A2**: Our theorems shows that the transition threshold $\eta = \sqrt{T/K}$ is sharp up to logarithmic factors. In fact, the two upper bounds in Theorem 4.1, $\tilde{O}(\sqrt{KT})$ and $\tilde{O}(\eta K)$ meet at $\eta = \sqrt{T/K}$ if we ignore all the logarithmic factors. Corollary 5.2 and Theorem 5.3 further confirm that in the corresponding regimes (i.e., $\eta\gtrsim\sqrt{T/K}$ and $\eta\lesssim\sqrt{T/K}$), the upper bounds are tight up to logarithmic factors. This indicates that $\eta = \sqrt{T/K}$ is the exact (up to logarithmic factors) transition threshold. We conjecture that the logarithmic factors in the *transition threshold* might be more accurately characterized through a fine-grained analysis. We leave a complete characterization of the threshold as future work.
>
> **Q3**: Is it possible to obtain instance-dependent regret bounds under the KL-regularized objective?
>
> **A3**: We would like to answer this question from two aspects. In the high regularization regime, the effect of regularization dominates, therefore the standard notion of instance hardness (e.g., $\sum_{a \in \mathcal{A}} 1/\Delta_a$) might not be applicable here and we need some other notion to characterize the hardness of a instance. On the other side, in the low regularization, the problem behaves similar to standard MABs, thus obtaining instance-dependent regret bounds is possible. We think this is an interesting direction and leave it as our future works.
>
> ---
>
> References:
>
> [1] Audibert, Jean-Yves, and Sébastien Bubeck. "Minimax policies for adversarial and stochastic bandits." Colt. 2009.
>
> [2] Zhao, Heyang, et al. "Logarithmic Regret for Online KL-Regularized Reinforcement Learning." International Conference on Machine Learning. PMLR, 2025.
>
> [3] Zhao, Qingyue, et al. "Towards a Sharp Analysis of Learning Offline $ f $-Divergence-Regularized Contextual Bandits." The Fourteenth International Conference on Learning Representations.

---

> > ### Author Rebuttal · Reviewer_Lbos · 2026-04-04
> >
> > I thank the authors for the detailed response and the additional experiments. The log $T$ gap remains open, but I accept this as a limitation rather than a flaw. I am satisfied with the rebuttal and maintain my positive score.

---

> > > ### Author Response · Authors · 2026-04-05
> > >
> > > Thank you for your support!

---

### Official Review · Reviewer_jhTT · 2026-03-12

**Soundness:** 2
**Presentation:** 3
**Significance:** 3
**Originality:** 3
**Overall Recommendation:** 5
**Confidence:** 5

**Summary:**

The paper considers the problem of regret minimization in KL-regularized multi-armed bandits, where the regularization coefficient is $\eta^{-1}$. The authors first prove the regret bound of $\tilde{O}(\eta K (\log T)^2 \wedge \sqrt{KT \log T})$ for KL-UCB of Zhao et al. (2025b), which is tighter by an order of $K$ compared to the prior guarantee by Zhao et al. (2025b). The main technical novelty is the peeling technique for bounding the sum of inverse visitation counts. The authors then prove a lower bound of the form $\Omega(KT \vee \eta K \log(T / (\eta^2 K)))$, which nearly matches the upper bound. Here, the main technical novelty is the construction of a Bayes prior over hard instances.

**Compliance With Llm Reviewing Policy:**

Affirmed.

**Final Justification:**

I have no further concerns. I have updated my score.

**Key Questions For Authors:**

**Key Questions**
1. Please respond to all points in Weakness 1. This is my main reason for not giving a higher score.

2. The lower bound proof relies on the fact that $\pi_{ref}$ is uniform, while the upper bound does not. Is there any chance for the lower bound proof to extend beyond uniform $\pi_{ref}$? If this is possible, then the lower bound is arguably much stronger, which would motivate me to consider a higher score.

3. The intuition for (poly)logarithmic regret seems that the gap condition is somewhat equivalent to the strong convexity induced by KL, although due to the KL-regularization here, there is no notion of "optimal arm". Would it be possible to interpolate from $\eta K (\log (K T))^2$ to $\sum_{i \neq i^\star} \frac{\log T}{\Delta_i}$ (when $\eta$ is sufficiently large, where $\Delta_i = r_{i^\star} - r_i$ is the suboptimality gap of the unregularized reward) to $\sqrt{K T \log (KT)}$ with the current algorithm?

4. The $\log T$ gap between upper and lower bounds is not discussed sufficiently. Considering how the logarithmic factors play a crucial role in the notion of regret optimality here, can the authors provide some explanations on whether the lower bound is not tight, or whether this is an artifact of the proof of the regret upper bound, or maybe a limitation of the current algorithm?



**Minor Questions**
1. In Remark 4.3, the authors mention that simply instantiating the general function approximation-based regret bound of Zhao et al. (2025b) is suboptimal in $K$. Can the authors elaborate on why this is the case?

2. Can the authors briefly elaborate on how the lower bound proof idea is related to prior works (Vovk, 2001; Singer et al., 2002; Zhao et al., 2023)? Like (very briefly and high-level-ishly), what were the prior works' techniques and how are they related to the construction in this paper?

3. Are there no requirements for $\pi_{ref}$ at all, e.g., does the regret upper bound hold for any arbitrary $\pi_{ref}$, even like a mixture of Diracs?


=====
(2026.04.01) After authors' responses, I have decided to raise my score from 4 to 5.

**Limitations:**

yes

**Strengths And Weaknesses:**

**Strengths**
1. Two new technical tools, potentially of interest: peeling technique for bounding sum of expected inverse visitations, and construction of Bayes prior over hard instances.
2. First regret lower bound for any KL-regularized bandits scenario, to the best of my knowledge. The lower proof itself is quite non-trivial.
3. The text makes a decent effort to clearly communicate the proof sketch and the related technical novelties.


**Weaknesses**
1. There are some (seemingly) wrong statements, both in the main text and in the proof:
    1. In the Introduction, the authors state that Wu et al. (2025) achieve $\tilde{O}(\exp(\eta) d_R \log (N_R) \log T)$ regret. But looking at their Appendix D.2, I think it should be $\tilde{O}(\eta \exp(2 \eta) d_R \log (N_R T / \delta))$.
    2. In Remark 5.4, the authors claim that a regret lower bound implies a sample complexity lower bound. I think this is wrong? By online-to-batch, it is true that regret *upper* bound implies sample complexity *upper* bound, but not vice-versa. Or if I'm wrong, please correct me, and elaborate more on how this is possible.
    3. In lines 418-422, the authors state that "Consequently, there exists $m = \Omega(K)$ arms for which the corresponding average KL-divergences are $O(1)$." First, I think $O(1)$ should be $\Omega(1)$. But more importantly, I struggle to understand how this is implied from the above averaging statement. For me, I think what the above implies is that there exists an arm $j \in [A]$ such that its averaged KL $\frac{1}{V} \sum_{\mu \sim_j \lambda} \mathrm{KL}_{\mu,\lambda} \geq 2$. How do you conclude that there exists $m$ such arms here?
    4. In Appendix B.2. in step 1 the authors say that the reward parameter $\bf{x}$ satisfies $|\bf{x}_i| \leq \alpha + \delta$ for all $i$'s. But then in line 979-980, the authors claim that "straightforward since we have $-\alpha \leq x_j \pm \delta \leq \alpha$. At least under the given premise, I don't think this is true...? Because $-\alpha \leq x_j + \delta \leq \alpha + 2 \delta$ and vice versa for $x_j - \delta$. Later, the authors then state "$\bf{x} \in [-\alpha + \delta_t, \alpha - \delta_t]^K", which seems to be inconsistent with the original premise. I don't think this is that much critical, but it still undermines the correctness of the proof.
        - Also here, please use $x_i$ instead of $\bf{x}_i$ when referring to the coordinates for notational clarity. This was a bit confusing.
    5. (Minor) In line 661, should $=$ be $\leq$?
    6. (Minor) In line 160, "$O(\log T)$ bound in the KL-regularized" => "$O((\log T)^2)$ bound in the KL-regularized"

2. Notation inconsistencies:
   - Throughout the authors use the notation $\tilde{O}$ and $\tilde{\Theta}$, but include $\log T$ dependencies. Looking at the proofs, I see that the other omitted factors are not too complex. Thus, I suggest that the authors just use $O$ notation and include all factors. In the end, the authors are claiming optimality w.r.t. logarithmic dependencies on $T$.

---

> ### Author Rebuttal · Authors · 2026-03-29
>
> Thank you for the thoughtful and constructive feedback! We focus on the most vital concerns due to space limit.
>
> **W1.1**: The upper bound in Wu et al. (2025a) is $\tilde O(\eta\exp(2\eta)d_{\mathcal{R}}\log(T N_\mathcal{R}))$.
>
> **A1.1**: Thank you for catching this slip! We will correct it in our revision.
>
> **W1.2**: Remark 5.4: A regret lower bound implies a sample complexity lower bound.
>
> **A1.2**: Thank you for catching this subtlety! We'll detail the mechanism in our revision. In short, while merely a regret bound cannot yield a sample complexity lower bound, such a sample complexity lower bound **can be derived from the proof** of Theorem 5.3: at line 1005, if we set $\delta=\sqrt{K/T}$, then recalling that $m=\Omega(K)$, we obtain $\text{SubOpt}_1(\hat \pi) + \text{SubOpt}_2(\hat \pi) \gtrsim \eta K T$. This yields the sample complexity lower bound of $\Omega(\eta K/\epsilon)$ stated in Remark 5.4.
>
> **W1.3**: In lines 418-422, the authors state that "Consequently, there exists $m=\Omega(K)$ arms for which the corresponding average KL-divergences are $O(1)$."
>
> **W1.3.1**: Here $O(1)$ should be $\Omega(1)$.
>
> **A1.3.1**: **This $O(1)$ is correct.** The general workhorse of proving lower bounds is the indistinguishability of two distributions, and such an indistinguishability leads to costs. The closeness of two distributions is usually characterized by their KL-divergence, where a small KL-divergence implies that two distribution are close. Thus, in order to show two distributions are close, we need their KL to be upper-bounded (e.g., $O(1)$). Actually, as pointed out in footnote 3, an $O(1)$ KL-divergence is enough.
>
> **W1.3.2**: How do you conclude that there exists $m$ such arms here?
>
> **A1.3.2**: Recalling that our goal is to bound the KL from above, we want enough arms on which the KL-divergence is upper bounded. It is routine to verify that $K$ positive numbers with $O(1)$ average implies $\Omega(K)$ of them are $O(1)$.
>
> **W1.4**: The issue on the range of $\mathbf{x}$ in the Proof of Theorem 5.3.
>
> **A1.4**: Sorry for two typos in Appendix B.2. In particular,  the premise $\\|\mathbf{x}\\|\_{\infty} \leq \alpha + \delta$ in line 904 and line 907 should be changed to $\\|\mathbf{x}\\|\_{\infty} \leq \alpha - \delta$, and on the 3rd line of the 1st equation block in Step 1 (line 914), it should read $r_1(i) = r_2(i) = 1/2 + x_i + r^\*(i), r^\*(i) \in \{\pm \delta\}, \forall i \in [m+1, K]$. After these two typo corrections, the logic of the proof will be correct because the configuration in Step 2 (i.e., the 1st paragraph of Step 2) always ensures $\alpha > \delta_t$ and $\\|\mathbf{x}\\|_{\infty} \leq \alpha - \delta_t$.
>
> **W1.5 & W2**: Minor typos; Expand out $\tilde O$ and $\tilde\Theta$; $\mathbf{x}_i$->$x_i$ for coordinates.
>
> **A1.5 & A2**: Thank you for pointing out these issues! We will fix them in our revision.
>
> **Q2**: Is it possible for the lower bound proof to extend beyond uniform $\pi_\text{ref}$?
>
> **A2**: All of our lower bounds are to show the worst-case hardness of the problem solved in Theorem 4.2. Thus, one set of hard instances in the proof **already verifies** the near-optimality of Theorem 4.2. Since our lower and upper bounds are nearly matched, a lower bound argument w/ non-uniform $\pi^{\text{ref}}$ will not make the statement much stronger. We leave developing lower bounds for general $\pi^{\text{ref}}$ as future work.
>
> **Q3**: Is it possible to interpolate from polylog(T) to $\sum_{i\neq i^*}\log T/\Delta_i$?
>
> **A3**: We speculate that this is possible because our regularized objective is close to the standard one at small $\eta$. We leave a thorough investigation as future work.
>
> **Q4**: On the $\log T$-gap between the upper and lower bounds in small $\eta$.
>
> **A4**: We refer to **W1 of Reviewer Lbos**.
>
>
> **Q5**: Why is simply instantiating the upper bound in Zhao et al. (2025b) suboptimal in $K$?
>
> **A5**: Zhao et al. (2025b) obtained a $\tilde{O}(\eta d_{\mathcal{R}}\log N_{\mathcal{R}}\log T)$ regret. When reduced to MABs, both the eluder dimension $d_\mathcal{R}$ of the function class, and $\log N_\mathcal{R}$, the log-covering number of the function class, are linear in $K$, resulting in a $K^2$ term.
>
>
> **Q6**: How is the lower bound proof idea related to prior works?
>
> **A6**: These related works construct continuous Bayesian prior distribution, in contrast to the categorical prior distribution in Le Cam or Assouad-type argument: the proof of lower bound in Vovk (2001) considers a uniform distribution over $[0,1]$. Singer et al. (2002) improved the result by modifying the prior to a Beta distribution over $[0,1]$. Zhao et al. (2023) employed a similar technique in an online regression setting.
>
> **Q7**: Does the regret upper bound hold for arbitrary $\pi_{\text{ref}}$?
>
> **A7**: Yes. If $\pi_\text{ref}$ is not fully supported on $\mathcal{A}$, the learner only needs to focus on $\text{supp}(\pi_\text{ref})$, which makes learning even easier.

---

> > ### Author Rebuttal · Reviewer_jhTT · 2026-04-01
> >
> > All my questions and concerns have been adequately addressed, and after reading through other responses, I have decided to raise my score from 4 to 5.
> >
> > I do have one follow-up question regarding the authors' rebuttal experiments:
> > - In Figure 2 of the anonymous link, it seems that there is no last iterate convergence towards $\pi^\star\_\{KL\}$ for large $\eta$: the TV distance seems to oscillate at around 0.3~0.4. Is this because the horizon is yet too small, or is it inherent? If it is inherent, then would online-to-batch conversion fix this? i.e., does the TV distance between $\pi^\star_\{KL\}$ and the uniformly average policy decay to zero? To me, the latter case makes sense, as, to my understanding, the submitted draft does not provide explicit "theoretical guarantees on policy convergence".

---

> > > ### Author Response · Authors · 2026-04-01
> > >
> > > We thank the reviewer for the detailed feedback and are glad that our response addressed the reviewer’s concern. We answer the follow-up question as follows.
> > >
> > > **Q1**: In Figure 2 of the anonymous link, it seems that there is no last iterate convergence towards $\pi^*_{\text{KL}}$ for large $\eta$: the TV distance seems to oscillate at around 0.3~0.4. Is this because the horizon is yet too small, or is it inherent?
> > >
> > > **A1**: We extend the time horizon to $T=10^6$ and re-plot Figure 2, the updated version of which is available [at this link](https://anonymous.4open.science/api/repo/KLMAB-449C/file/Rebuttal.pdf). The experimental results show that, while the oscillation persists around $t=10^3$ to $t=10^4$, the  trend as $t$ approaches $10^6$ suggests that $\pi_t$ converges to $\pi^*_{\text{KL}}$ in TV distance. Nevertheless, the primary focus of this paper is KL-regularized regret minimization, and we agree with the reviewer that our submission does not provide theoretical guarantees for policy convergence. Therefore, the convergence behavior suggested by the empirical results remains conjectural. We leave a thorough investigation of policy convergence as our future work.

---

### Official Review · Reviewer_jjPo · 2026-03-15

**Soundness:** 3
**Presentation:** 3
**Significance:** 3
**Originality:** 3
**Overall Recommendation:** 4
**Confidence:** 3

**Summary:**

The paper studies online learning in multi-armed bandits with KL-regularized objectives, with the goal of clarifying the statistical complexity of this setting across different regularization regimes. The main technical contribution is a high-probability analysis of KL-UCB, based on a new peeling argument, which yields an upper bound of order $\widetilde{O}(\eta K \log^2 T)$, in particular with linear dependence on the number of arms (K). The paper also establishes a lower bound of order $\Omega(\eta K \log T)$, which is claimed to be the first non-constant lower bound for this problem. In addition, for the low-regularization regime, the authors show that the KL-regularized regret becomes independent of (\eta) and scales as $\widetilde{\Theta}(\sqrt{KT})$. Overall, the paper aims to provide a nearly complete characterization of the dependence of KL-regularized bandit regret on $K, \eta$, and $T$.

**Compliance With Llm Reviewing Policy:**

Affirmed.

**Final Justification:**

Final justification:

The rebuttal successfully addressed my main questions, which were primarily about presentation and positioning rather than about the technical correctness of the results. In particular, I appreciate the authors’ clarification that KL-regularized regret should be explicitly distinguished from standard cumulative regret, and I found their explanation of the relationship between the two notions useful. The rebuttal also clarified why prior work involves more complicated regret expressions, namely because those works study richer settings such as contextual bandits, linear or general function approximation, and associated coverage or complexity parameters. This strengthens the paper’s positioning as a sharp characterization in the minimalist multi-armed bandit setting.

Overall, the rebuttal resolved my main concerns, but these were mostly requests for clarification rather than fundamental objections. As a result, my overall evaluation remains unchanged. I continue to view the paper as a technically solid and meaningful contribution, and I keep my original score.

**Key Questions For Authors:**

Please check the 'Strength and Weakness' section for details.

**Strengths And Weaknesses:**

The analysis in the paper itself appears solid, and it is certainly true that KL-regularization has attracted substantial attention in reinforcement learning. In that sense, bringing KL-regularized objectives into bandits, as a more specialized RL setting, is clearly meaningful and interesting. The algorithm is also not drastically different from standard UCB-type methods, and it is noteworthy that the authors obtain a clear improvement over Zhao et al. (2025) through a delicate peeling argument. I also appreciated that the paper includes a careful lower-bound analysis, which helps support the claim that the result is near-optimal.

That said, the main weakness I would like to point out, and also the question I most want to ask, is the following:

1. Is KL-regularized regret really compatible with the usual notion of cumulative regret?

1-1) At several points in the paper—for example, in the “Optimism in MAB” part of the related work section—the discussion seems to mention cumulative regret alongside KL-regularized regret, and this at times made it unclear whether the two should be understood as essentially the same type of performance criterion. To me, the two notions of regret appear clearly different. For that reason, I think it would be better if the authors explicitly wrote “KL-regularized regret” already in the abstract and contribution statements, so as to avoid possible confusion.

1-2) More importantly, it would be very helpful if the paper could explain, in a way that is accessible to a broader bandit audience, how KL-regularized regret relates to the standard notion of cumulative regret. The paper does mention that in the large-(\eta) regime the result becomes closer to the standard MAB setting. However, this naturally raises the following question: when (\eta) is not large, a bandit algorithm may achieve logarithmic KL-regularized regret, but what then can be said about its ordinary cumulative regret?

2. The notation in prior work on KL-regularized bandits seems to vary substantially. Could the authors clarify, for all of the papers listed in Table 1, why this apparent complication arises? UCB combined with a peeling technique is certainly a strong and well-executed approach, but at least at first glance it also seems like a fairly natural approach if one is studying the same underlying problem. This makes the variety and complexity of the resulting regret expressions somewhat surprising. I would therefore appreciate some clarification as to whether those previous papers were in fact treating settings more general than bandits, for example broader RL formulations.

Overall, I find the paper sufficiently well executed and meaningful to merit a weak accept.

---

> ### Author Rebuttal · Authors · 2026-03-29
>
> Thank you for your supportive feedback! We answer your questions as follows.
>
> **Q1**: The two notions of regret appear clearly different. It would be better to explicitly wrote “KL-regularized regret” to avoid possible confusion.
>
> **A1**: We agree with the reviewer that the two types of regret notions are different and sincerely appreciate your suggestion. We will change our terminology to "KL-regularized regret" in the revision to avoid possible confusion.
>
> **Q2**: How KL-regularized regret relates to the standard notion of cumulative regret: when $\eta$ is not large, what can be said about its ordinary cumulative regret?
>
> **A2**: In particular, the KL-regularized objective $J(\pi) = \mathbb{E}\_{a \sim \pi} [r(a)] - \eta^{-1} \text{KL}(\pi \\| \pi^{\text{ref}})$ is composed of two terms, indicating a tradeoff between standard objective $\mathbb{E}\_{a \sim \pi} [r(a)]$ and the KL-divergence between $\pi$ and $\pi^{\text{ref}}$. Therefore, when $\eta$ is large, the effect of KL-term is small and therefore the objective (regret) is close to standard objective (regret). On the other side, when $\eta$ is small, the objective emphasizes $\pi$ being close to $\pi^{\text{ref}}$, resulting in a large difference from standard objective and regret. Consequently, an algorithm achieving logarithmic KL-regret might not achieve optimal ordinary cumulative regret. We will add this discussion to our paper in the revision.
>
> **Q3**: Why do the relevant previous works have involved and diverse notions and notations? Were they treating more general settings?
>
> **A3**: While all related results in Table 1 are for (contextual) (dueling) bandits, the complication of notations arises from the reward function class they considered in their problems. In particular, [1] considered linear function approximation, which introduces the dimension of the feature map $d$ as a model parameter. [2,3,4] consider general function approximation, thus introduce the problem parameters $d_{\mathcal{R}}$ and $\log N_{\mathcal{R}}$, which are the eluder dimension and the log-covering number of the function class respectively. The algorithm in [2] only samples from $\pi^{\text{ref}}$, which naturally introduces a dependency of data coverage $C_{\text{GL}}$ which characterizes the coverage of the behavior policy $\pi^{\text{ref}}$.
>
> While these setups incorporate multi-armed bandits, these results are not tight when specialized to MABs and all these results do not match the corresponding lower bound in the original setting. This motivates this work, which sets up the nearly matching upper and lower bounds in multi-armed bandits, which is minimalist setting of online learning.
>
> ---
>
> References:
>
> [1] Xiong, Wei, et al. "Iterative preference learning from human feedback: Bridging theory and practice for rlhf under kl-constraint." arXiv preprint arXiv:2312.11456 (2023).
>
> [2] Zhao, Heyang, et al. "Sharp analysis for kl-regularized contextual bandits and rlhf." arXiv preprint arXiv:2411.04625 (2024).
>
> [3] Zhao, Heyang, et al. "Logarithmic Regret for Online KL-Regularized Reinforcement Learning." International Conference on Machine Learning. PMLR, 2025.
>
> [4] Wu, Di, et al. "Greedy Sampling Is Provably Efficient For RLHF." The Thirty-ninth Annual Conference on Neural Information Processing Systems.

---

> > ### Author Rebuttal · Reviewer_jjPo · 2026-04-05
> >
> > Thank you for the helpful rebuttal. The authors clearly addressed my main presentation concern, and I appreciate their agreement that the terminology should consistently distinguish KL-regularized regret from standard cumulative regret. I also found the added explanation of the relationship between KL-regularized regret and ordinary cumulative regret useful: in particular, the rebuttal makes clear that when $\eta$ is small, logarithmic KL-regularized regret does not necessarily imply comparably strong guarantees for the usual cumulative regret.
> >
> > The clarification regarding prior work was also helpful. I now better understand that the more complicated regret expressions in earlier papers arise largely from the richer model classes they study, such as linear or general function approximation, contextual settings, and related coverage or complexity parameters, rather than from superficial notational differences alone. This strengthens the paper’s positioning as a sharp characterization in the minimalist MAB setting.
> >
> > Overall, the rebuttal resolves my main questions, but these were mostly requests for clarification rather than fundamental objections. As such, my overall evaluation remains the same, and I will keep my score unchanged.

---

> > > ### Author Response · Authors · 2026-04-05
> > >
> > > We are happy that our clarifications addressed your questions. We will incorporate your suggestions in the revision. Thank you for your support.

---

### Decision · Program_Chairs · 2026-04-30

**Decision:**

Accept (regular)

**Comment:**

The paper studies stochastic multi-armed bandits under a KL-regularized objective and provides a near-complete regret characterization across regularization regimes. Reviewers broadly agreed that the main contribution is not a new algorithmic template, but a sharp theoretical analysis: in particular, the improved linear dependence on the number of arms in the strongly regularized regime, together with substantive lower bounds, was viewed as a meaningful advance. The technical development was generally considered strong, with the peeling-based upper-bound argument and the hard-instance construction for the lower bound standing out as key strengths.

The main concerns were about presentation, statement precision, and positioning rather than the core correctness of the results. These included clarifying the distinction between KL-regularized regret and standard cumulative regret, sharpening some theorem statements, and fixing a number of typos and minor proof-level inconsistencies. The rebuttal addressed these points well, and in at least two cases led reviewers to strengthen their assessment. Overall, the paper appears technically sound, novel in its analysis, and significant within its scope, and I recommend acceptance.